

# Grazing enhances carbon cycling, but reduces methane emission in the Siberian Pleistocene Park tundra site

Wolfgang Fischer[1], Christoph K. Thomas[1,2], Nikita Zimov[3], and Mathias Göckede[4]

[1]Micrometeorology Group, University of Bayreuth, Bayreuth, Germany
[2]Bayreuth Center of Ecology and Environmental Research, BayCEER, Bayreuth, Germany
[3]North-East Science Station, Pacific Institute for Geography, Far-Eastern Branch of Russian Academy of Science, Chersky, Republic of Sakha (Yakutia), Russia
[4]Department of Biogeochemical Signals, Max Planck Institute for Biogeochemistry, Jena, Germany

**Correspondence:** Mathias Göckede (mgoeck@bgc-jena.mpg.de)

**Abstract.** Large herbivore grazing has been shown to substantially alter tundra soil and vegetation properties as well as carbon fluxes, yet observational evidence to quantify the impact of herbivore introduction into Arctic permafrost ecosystems remains sparse. In this study we investigated growing season $CO_2$ and $CH_4$ fluxes with flux chambers on a former wet tussock tundra inside Pleistocene Park, a landscape experiment in Northeast Siberia with a 22 year history of grazing. Reference data for an undisturbed system were collected on a nearby ungrazed tussock tundra. Linked to a reduction in soil moisture, topsoil temperatures at the grazed site reacted one order of magnitude faster to changes in air temperatures compared to the ungrazed site and were significantly higher, while the difference strongly decreased with depth. Overall, both $GPP$ (gross primary productivity, i.e. $CO_2$ uptake by photosynthesis) and $R_{eco}$ (ecosystem respiration, i.e. $CO_2$ release from the ecosystem) were significantly higher at the grazed site with notable variations across plots at each site. The increases in $CO_2$ component fluxes largely compensated each other, leaving $NEE$ (net ecosystem exchange) similar across grazed and ungrazed sites for the observation period. Soil moisture and $CH_4$ fluxes at the grazed site decreased over the observation period, while in contrast the constantly water-logged soils at the ungrazed site kept $CH_4$ fluxes at significantly higher levels. Our results indicate that grazing of large herbivores promotes topsoil warming and drying, effectively accelerating $CO_2$ turnover while decreasing methane emissions. Our experiment did not include autumn and winter fluxes, and thus no inferences can be made for the annual $NEE$ and $CH_4$ budgets at tundra ecosystems.

## 1 Introduction

In the context of global climate change, surface air temperatures in polar regions have been shown to rise about twice as fast as the global mean in the past (Overland et al., 2015). Since this trend is expected to continue in the future, Northern hemisphere permafrost ecosystems are at an exceptional risk for degradation. The Arctic permafrost region stores about 50% of the belowground organic carbon stocks on Earth (Hugelius et al., 2014), with an estimated pool of organic C between 1307 Gt and 1672 Gt (Hugelius et al., 2014; Tarnocai et al., 2009). Based on expert assessment, 130 to 160 Gt C could be released by 2100 under a high warming scenario (i.e., Representative Concentration Pathway scenario 8.5) (Schuur et al., 2015), so even





a partial release of this currently deep-frozen material would constitute a substantial positive feedback with ongoing warming trends.

Warming of the active layer facilitates the enhanced decomposition of soil organic carbon, leading to higher rates of ecosystem respiration ($R_{eco}$) that contribute to higher $CO_2$ emissions (Natali et al., 2015; Schädel et al., 2016; Schuur et al., 2009). Moreover, warming-induced permafrost thaw may make organic matter vulnerable to mineralization which was previously perennially frozen (Natali et al., 2014, 2015; Harden et al., 2012; Schuur et al., 2009). At the same time, warmer and longer growing seasons hold the potential to change vegetation species composition (e.g. from graminoid/moss- to shrub-dominated)
(Myers-Smith et al., 2011; Hollister et al., 2015) and increase living plant biomass, leading to an increase in gross primary productivity ($GPP$) (Epstein et al., 2012). Deeper thaw depths may also open up nutrient reservoirs (Chapin et al., 2005; Salmon et al., 2016; Hollister et al., 2015), this way promoting vegetation growth and carbon uptake. Interactions with various other influencing factors, for example changes in snow cover (Grogan, 2012) or soil moisture (Oberbauer et al., 2007; Natali et al., 2015; Kwon et al., 2016), further complicate an assessment of the net effect of these changes.

Large herbivores (i.e. reindeer, muskoxen, horses, bison etc.) are an additional forcing which may substantially alter the characteristics of high-latitude landscapes, but their potential influence is rarely considered in studies predicting the future state of Arctic permafrost ecosystems. Herbivores can trigger distinct shifts in vegetation communities, e.g. from shrub or moss-dominated ecosystems to graminoid tundra dominated by dense grass tillers (Manseau et al., 1996; Olofsson, 2006; Ylänne et al., 2018; Kitti et al., 2009; Falk et al., 2015; Raillard and Svoboda, 2000; Gornall et al., 2009). Grazing has been shown
to promote certain *Carex* species that produce a high belowground biomass (Tolvanen and Henry, 2000), allowing to reliably compensate growth after being grazed off (Raillard and Svoboda, 1999; Kitti et al., 2009). Shifts in vegetation composition are usually associated with an increase in albedo (Te Beest et al., 2016; Chapin et al., 2005; Cohen et al., 2013). Regarding belowground ecosystem properties, previous studies reported an increased annual amplitude in soil temperatures (higher in summer, lower in winter), and significant shifts in soil moisture and texture (Olofsson et al., 2004; Te Beest et al., 2016; Zimov
et al., 1995, 2012; Beer et al., 2020; Olofsson et al., 2001), which in combination with snow trampling in winter (Beer et al., 2020) tend to reduce annual permafrost temperatures. Finally, herbivore grazing can cause an increase in nutrient availability, and acceleration in nutrient cycling (Olofsson et al., 2004, 2001; Raillard and Svoboda, 1999, 2000). The combinations of these changes leads to strong and variable alterations in carbon cycle processes (higher $R_{eco}$, higher/lower $GPP$, higher/lower $NEE$) (Falk et al., 2015; Metcalfe and Olofsson, 2015; Väisänen et al., 2014; Cahoon et al., 2012; Ylänne et al., 2018; Ylänne
and Stark, 2019), with net effects highly dependent on site-specific characteristics.

    Most existing studies focusing on grazing effects were conducted in Scandinavian upland tundra, while other Arctic domains, and particularly wet tundra ecosystems (e.g., Falk et al., 2015; Raillard and Svoboda, 2000; Kitti et al., 2009), remain sparsely examined to date. Moreover, concerning carbon cycling only the effect in the growing season was investigated so far, despite the importance of non-growing-season fluxes in the tundra (Grogan, 2012; Euskirchen et al., 2012; Kittler et al., 2017).
A long-term, landscape-scale experiment called "Pleistocene Park", established in northeastern Siberia in 1996, provides an opportunity to address this research gap. While initially designed to reestablish a "mammoth steppe" ecosystem which dominated this region in the Pleistocene (Zimov et al., 2012), the experiment also facilitates to study the impact of large herds





of herbivores as an agent to stabilize permafrost ecosystems against degradation linked to Arctic warming. The underlying hypotheses of Pleistocene Park are that herbivore grazing a) increases carbon sequestration by simultaneously increasing pro-

ductivity and root formation during the growing season, b) decreases annual permafrost temperatures by trampling the snow in the winter, thereby limiting permafrost thaw and respiration, c) increases the surface albedo by decreasing shrub and tree cover, and d) decreases $CH_4$ emissions by decreasing soil moisture through condensing soils and increased evapotranspiration by a more active vegetation.

While all the grazing effects listed above hold the potential to reduce future positive feedbacks between alterations of the

permafrost carbon cycling and the changing climate, so far only limited observational evidence was presented allowing for evaluating the management effects inside Pleistocene Park. The main objective of the presented study is therefore to provide new insights into the effects of herbivore grazing on carbon cycle processes and ecosystem characteristics within the park. For this purpose, we compare growing season carbon fluxes from flux chamber measurements along with soil parameters and radiation balance components across an intensively grazed area within Pleistocene Park and a nearby undisturbed site.

## 2 Materials and Methods

### 2.1 Site Description

The study area is located in the Kolyma Lowlands region in northeastern Siberia (68.51°N, 161.50°E), close to the town of Chersky, Sakha Republic, Russia, which is situated around 100 km south of the Arctic Ocean. The weather patterns frequently switch between maritime air masses from the North and continental air masses from the Southeast, with the former dominating

the wintertime conditions, and the latter in the summer. The mean daily air temperature can remain at or below -40°C for several days during December to February, while daily means of more than 20°C can be reached around the peak of summer. The mean annual temperature (averaged for 1960-2009) is approximately -11.0°C . The total amount of annual precipitation (averaged for 1950-1999) is between 200 and 215mm, with 80 to 110mm falling as rain (Göckede et al., 2017). Snow-melt leads to an annual flooding event in the Kolyma River and its tributaries, usually inundating large parts of the study area between mid to

late May and late June (Kwon et al., 2016).

In the context of this study, we compared carbon fluxes and ecosystem characteristics between two measurement sites in the Chersky region. The first one, Pleistocene Park, hosts a variety of herbivores (sheep, yaks, cows, horses, bison, muskoxen, reindeer), and was used to study the effects of grazing on permafrost ecosystems. These observations were compared to those at a second, non-grazed tussock tundra site several kilometers from the park as a reference.

The Pleistocene Park area (https://pleistocenepark.ru) was established in 1996 on 144 $km^2$ of permafrost territory mainly consisting of ecotonal upland forest-tundra, wet lowland tundra, and small lakes and rivers. About 2000 hectares have been fenced in to concentrate animals on the core domain of the park. Inside the wet lowland section, we selected one of the longest and most intensively grazed areas as our grazing study site hereafter labeled as "GR" with data being collected at three plots (GR-1 to GR-3). This site, which gets flooded every year in spring during snow-melt, is a moist-wet meadow almost

without shrubs, also featuring decaying tussocks. The vegetation at this site primarily consists of grasses and sedges, including



*Calamagrostis langsdorfii*, *Carex appendiculata*, and *Eriophorum spec.* (Euskirchen et al., 2017). Before the introduction of grazing herbivores, this site used to be dominated by tussocks and saturated with water during the whole year (Sergey Zimov, pers. comm.). It therefore represents lands disturbed by grazing representative for the lowland wet tussock tundra dominating large parts of the Kolyma Lowlands region.

The reference site, hereafter labeled as ungrazed site "UGR", is located outside the Pleistocene Park domain on a wet-tussock tundra floodplain along the Ambolikha river, a small tributary of the Kolyma river. Here, a long-term monitoring site was installed to investigate drainage effects on wet tundra ecosystems (Kwon et al., 2016, 2017; Göckede et al., 2017). The reference site for our study is the non-drained control area of this experiment. The dominant vegetation species are tussock-forming *Carex appendiculata* and *lugens*, and *Eriophorum angustifolium*, with betula nana and willow spec. growing on elevated areas with a lower water table. An organic peat layer (15–20 cm deep) has accumulated on top of alluvial material soils (silty clay) (Kwon et al., 2017). In the context of this study, we assume this site to reflect the status that the grazed site would have if herbivores had not been introduced there, since both sites showed a similar ecosystem structure in the early 1990s (Sergey Zimov, pers. comm.). Observations were collected at two plots (UGR-1, UGR-2) to capture some variability concerning vegetation structure and soil properties. Based on results of a previous study evaluating small-scale flux variability across a transect of ten quasi-randomly selected locations (Kwon et al., 2016), flux rates and environmental conditions within these two plots were shown to be close to the average at this site.

## 2.2   Measuring Radiation and Soil Parameters

At both GR and UGR sites, soil temperature probes (model Th3-s Soil Temperature Profile Probe, UMS GmbH Munich, Germany) were installed, measuring soil temperatures at 5cm, 15cm, 25cm and 35cm depth. For soil moisture (SM) measurements, we installed three TDR probes (time-domain reflectometry; models CS 640, 630, and 605, Campbell Scientific, Logan, UT, USA) at depths of 7.5cm, 15cm and 30cm close to the soil temperature sensor. At the UGR site, due to the water-logged conditions, soil moisture was permanently saturated, and no measurements were used within the context of this study.

SM values at GR were flagged using plausibility limits, and systematic offsets corrected. Since soil data were only collected during times of flux measurements, values were subsequently interpolated based on e.g. air temperatures and incoming radiation to create continuous time series.

Radiation budget components in the Pleistocene Park were measured using a net radiometer (model a CNR1, Campbell Sci., Logan, UT, USA) installed at 4 m height on a pole approximately 15 m away from the flux sampling sites. At the Ambolikha site, a net radiometer (model CNR4, Campbell Sci., Logan, UT, USA) permanently installed on a 5 m tall flux tower located approximately 50 m away from the sampling sites provided radiation observations. Measurements were stored as 10 minute averaged intervals. Albedo was derived by dividing the average upwelling shortwave radiation by the average downwelling component for each single day. Then, these daily averages were averaged over the observation period. While photosynthetically active radiation (PAR) was measured directly at the UGR site, at GR it was converted from incoming shortwave radiation data based on the approach by Britton and Dodd (1976).





## 2.3 Measuring Fluxes using Chambers

Directly prior to measurements at GR, wooden fences were constructed to protect the sites from grazing animals during chamber operation. At both GR and UGR sites, walking boards were placed around the setup to prevent damaging plants and minimize influences on measurements by disturbing the soil. Our flux chamber approach closely followed Kwon et al. (2016, 2017), and is therefore only briefly outlined in the following paragraphs.

Carbon dioxide, $CO_2$, and methane, $CH_4$, fluxes were determined with a non-steady-state flow-through method using an
Ultra-Portable Greenhouse Gas Analyzer (UGGA, Los Gatos Research, USA) for in-situ measurements of gas concentrations at 1 Hz. 60cm*60cm PVC collars, which have a socket at the top for the chamber, were installed in the ground at each plot to prevent leaking of air during chamber measurements. The cubic chamber hoods with 60 cm side length, made of 4 mm thick plexiglass, were placed on these collars to capture gases exchanged with the surface. They featured an opening valve on the top to avoid pressure effects when the chamber is placed onto the collars. Inside the chamber, three electric fans were
installed to ensure well-mixed conditions. Air was pumped from the chamber to the gas analyzer through three tubes installed at different heights inside the chamber hood. Sensors for air temperature ($T_{air}$), relative air humidity ($rH$), air pressure ($P_{air}$), and photosynthetically active radiation ($PAR$) were attached to one side of the chamber. For measurements, the chamber was oriented in a way to minimize shading the vegetation with the instruments.

Each flux measurement was restricted to a maximum of two minutes in order to minimize disturbance effects such as e.g.
temperature increases or moisture saturation within the chamber (e.g., Kutzbach et al., 2007). After completing one measurement, the chamber was lifted and tilted for ventilation until ambient $CO_2$ concentrations were reached. Ecosystem respiration ($R_{eco}$) was determined by covering the hood with a white polyethylene tarp that completely blocked incoming radiation. For each plot, one measurement iteration consisted of three NEE measurements and two $R_{eco}$ measurements.

**Table 1.** Number of utilizable light ($NEE$) and dark ($R_{eco}$) measurements for each chamber site and total number of measurement days.

|       | UGR-1 | UGR-2 | GR-1 | GR-2 | GR-3 |
|-------|-------|-------|------|------|------|
| light | 77    | 40    | 77   | 71   | 68   |
| dark  | 45    | 27    | 46   | 42   | 41   |
| days  | 4     | 4     | 9    | 9    | 9    |

On each sampling day at GR, we rotated between chamber locations, with about one full hour needed to complete the five
individual measurements at each of the three plots. Since the two plots at UGR were located further apart, we conducted several individual measurements at one location before switching to the other to minimize time needed for relocating instrumentation. On each day, only one of the study sites (UGR vs. GR) was sampled. The total quantity of measurements is shown in Tab.1.

## 2.4 Calculation and Interpolation of Carbon Fluxes

Each chamber measurement resulted in a 1 Hz time series of $CO_2$ and $CH_4$ concentrations. After excluding the equilibration
period of at least 5-10 seconds, periods with a steady concentration change in greenhouse gases were identified, and an ensem-





ble of stationarities (slopes) were calculated for varying start and end times within this period using a bootstrapping approach. The final slope used for flux computations was identified as the frequency distribution's median. Implausible or disturbed signals were manually flagged, and excluded from further analysis. Such cases included for example unstable signals without a distinctly discernible, steady slope, which are not clearly interpretable, or signals obviously disturbed by leakage.

The median slope ($\widehat{a}$) of greenhouse gas concentrations change over time were transformed to a flux using the following formula:

$$Flux = \widehat{a} \frac{\frac{V_{ch}}{A_{ch}} p_{air}}{R T_{air}} \tag{1}$$

$V_{ch}$ and $A_{ch}$ are the volume and surface area of the chamber, respectively. $R$ is the ideal gas constant ($8.3144 J/molkg$), $T_{air}$ and $p_{air}$ are the mean air temperature (K) and pressure (Pa) inside the chosen interval. Fluxes are derived in units of

$[\mu mol C - CO_2 m^{-2} s^{-1}]$ and $[\mu mol C - CH_4 m^{-2} s^{-1}]$, respectively.

While fluxes of $R_{eco}$ and $NEE$ could be calculated based on dark and light chamber measurements, respectively, the photosynthetic uptake portion of the flux ($GPP$) was calculated as the difference between measured $NEE$ and the mean of measured $R_{eco}$ for one measurement. The standard error (RMSE) of each flux measurement was calculated using all bootstrapped slopes, distinguishing between $R_{eco}$ and $NEE$ measurements. The slope error for $GPP$ was taken as the summed

errors of $R_{eco}$ and $NEE$ measurements. Error values are given in Tab. A1 in the Appendix. Calculations were conducted using the software package R-studio.

To analyze the implications of the grazing disturbance on net $CO_2$ and $CH_4$ exchange across the sites, the time series of flux estimates for each plot was interpolated across the entire measurement period of 17 days at a resolution of 10 minutes. Both $R_{eco}$ and $CH_4$ fluxes were interpolated using empirical plot-specific linear models linking flux rates to environmental

parameters ($T_S$, $T_{air}$, and $SM$). Parameters minimizing the fit uncertainty were not uniform across plots even at a site (see Appendix A for an exact description of this approach, including the derivation of formulas, and chapter 4.2. for evaluation). The following formulas were derived to interpolate $R_{eco}$ and $CH_4$ fluxes:

$$R_{eco}(GR-1, GR-2) = exp(a_0 T_{Soil,5cm} + b_0) + a_1 SM_{7.5cm} + b_1 \tag{2}$$

$$R_{eco}(GR-3, UGR-1, UGR-2) = exp(a_0 T_{air} + b_0) \tag{3}$$

$$F_{CH_4}(UGR) = exp(a_0 T_{Soil,15cm} + b_0) \tag{4}$$

$$F_{CH_4}(GR) = a_0 T_{Soil,25cm} + b_0 + a_1 SM_{15cm} + b_1 \tag{5}$$





In each formula, $a_0$ is the slope of the first applied model, $a_1$ for the second. $b_0$ and $b_1$ are corresponding intercepts. Total errors for all fluxes were derived considering the standard error from the final model compared to observed values (linear regression), further considering the standard error from the bootstrapping approach used to transfer measured concentration slopes into fluxes and the standard error from modeled $T_S$ (for $R_{eco}$ and $CH_4$ fluxes). A detailed error calculation is shown in Appendix A3.

GPP was modelled as a function of PAR, using a rectangular hyperbolic function (Runkle et al., 2013).

$$GPP = -\frac{P_{max}\alpha PAR}{P_{max} + \alpha PAR} \tag{6}$$

The fit parameters $\alpha$ and $P_{max}$ represent, respectively, the initial canopy quantum efficiency (the initial slope of the GPP-PAR
curve at PAR= 0) and the maximum canopy photosynthetic potential, which is the hypothetical maximum of GPP at infinite PAR. Both $\alpha$ and $P_{max}$ are assumed to have positive values, necessitating the negative sign on the equation's right-hand side to allow GPP to fit the NEE sign convention. Hereby, positive fluxes imply carbon losses from the ecosystem into the atmosphere. This model contains the explicit assumption that GPP is insensitive to light stress or temperature effects (Runkle et al., 2013). For each site, $\alpha$ and $P_{max}$ were determined by fitting PAR against the $GPP$-Fluxes from chamber measurements and applying
a non-linear-least-squares (nls) optimization. Implausible PAR values from chamber measurements were replaced by PAR derived from the net radiometer measurements. With these parameters, a continuous $GPP$ time series could be modeled for the entire observation period.

### 2.5 Statistics

To visualize and compare carbon fluxes between plots and study sites, daily means for $GPP$, $R_{eco}$, $NEE$ and $CH_4$-fluxes
were calculated. Daily average $T_S$ and carbon fluxes were compared using a two-sided Mann-Whitney-test (?) in R. Mean daily albedo and $R_{net}$ were compared using a two-sample t-test.

## 3   Results

### 3.1   Environmental conditions and ecosystem characteristics

#### 3.1.1   Albedo and energy fluxes

Linked to the differences in vegetation community structure described in Section 2, the average daily albedo was significantly higher at GR, with an average value of 0.217, compared to the UGR (0.192, $p < 0.0001$). This enhanced reflectivity of the surface increased the upwards directed shortwave radiation by an average of 6.3 $Wm^{-2}$ over the course of the observation





period (mean $SW_u$ GR: 48.6 $Wm^{-2}$ ; UGR: 42.3 $Wm^{-2}$), in effect reducing the net radiative energy available to the grazed
ecosystem.

### 3.1.2 Soil moisture

Soil hydrologic conditions were distinctly different between grazed and ungrazed sites. At the beginning of the observation
period, both sites were primarily water-logged since the water levels from the preceding spring flooding had not fully receded.
From this starting point, we observed deviating temporal dynamics in drying between both ecosystems during the peak of the
growing season. Water levels declined only marginally, or not at all, within the ungrazed reference area, with water levels re-
maining above ground. Soil moisture at the grazed ecosystem (GR), on the other hand, gradually decreased across all measured
depths over the period of observation, especially in the topsoil ($SM_{7.5cm}$: 63.1% - 49.9%; $SM_{15cm}$: 64.3% - 54.2%; ,$SM_{30cm}$:
60.7% - 56.7%, changes from the first to the last measurement day, respectively).

### 3.1.3 Air and soil temperatures

A change in the general weather pattern around mid July 2019 split our observation period roughly into a warm and sunny
first half, and a cool and cloudy second half. Regarding the mean air temperatures, during the first period (July 07 - 15), daily
average $T_{air}$ ranged between 14.3 - 26.9°C, while conditions during the second period (July 16 - 22) were much cooler (6.9 -
10.9°C). Mean air temperatures at the grazed site were observed to be about 1°C warmer compared to the ungrazed site, with
similar daily minima, while daily maxima were distinctly higher at GR.

The trends in soil temperatures matched those described for air temperature: while $T_{soil}$ across the observed vertical profile
were rising during the first part of the observation period, in the second half they declined (see also Fig. 1). We found topsoil
temperatures in 5 cm depth at the grazed site (GR: max 19.6°C, min 9.0°C) to be significantly higher ($p < 0.0001$) compared
to the ungrazed reference (UGR-1: max 15.4°C, min 5.6°C; UGR-2: max 11.6°C, min: 4.3°C) during the whole observation
period. The time lag of $T_{soil}$ reacting to $T_{air}$ was one order of magnitude shorter at GR - moving averages of air temperatures
explaining $T_{soil}$ in 5 cm integrated the last 4.3h ($p < 0.0001$) at GR, while reaching back 40.7h($p < 0.0001$) at UGR-1 and
86.7h ($p < 0.0001$) at UGR-2. However, this difference vanished for $T_{soil}$ in 15 cm depth (100h, 67.5h, 108.3h at GR, UGR-
1 and UGR-2, respectively; $p < 0.0001$). No comparisons could be made for $T_{soil}$ in 25 cm and 35 cm depth, respectively,
because no significant correlation was found due to limited data availability at GR.

    In the deeper soil layers (15 cm, 25 cm, 35 cm), the temperatures at the UGR-2 site were consistently lower than observed at
all other sites (GR, UGR-1). Comparing observations from GR and UGR-1, during the first week sites showed similar average
soil temperatures, while during the second week soils became clearly warmer for the drier grazed site. Notably, $T_{soil}$ in 35
cm and 25 cm at GR were lower compared to UGR-1 during the first measurement days, but due to a steeper warming rate
at the grazed site this changed after 5 days into the observations. In the second week, $T_{soil}$ in 35 cm and 25 cm at GR were
significantly higher compared to UGR-1, most pronounced in 35 cm depth ($p < 0.0001$).

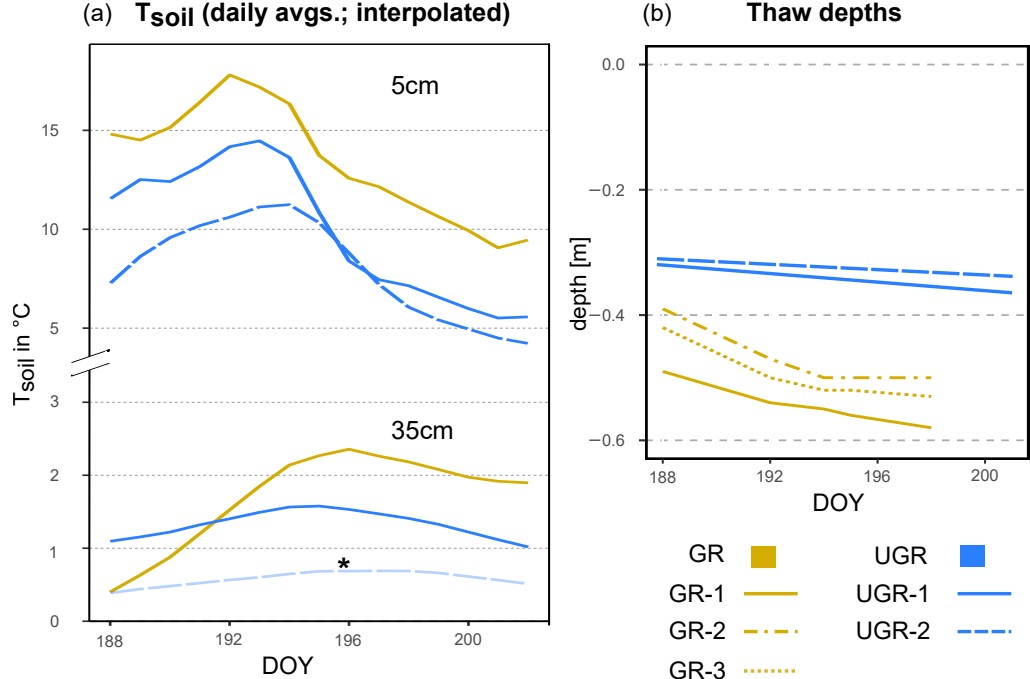

**Figure 1.** (a) Soil temperature ($T_{soil}$) at GR, UGR-1 and UGR-2 in 5 cm and 35 cm depth. The interpolation model of $T_{soil}$ in 35 cm at UGR-2 (*) is not based on a significant fit, but follows the expected course closely, and is therefore included here for overview. (b) Evolution of thaw depths at all plots.

### 3.1.4 Thaw depths

Measured thaw depths were greater at all GR sites compared to all UGR sites throughout the observation period: while the values within the ungrazed reference area varied between 31 - 36 cm over time and across sites, that range was 39 - 58 cm at the grazed site. Importantly, also the temporal dynamics as well as the variability across sites differed strongly between these two study areas. At UGR, the average increase in thaw depths was 0.25 cm per day during the observation period. Even though both observation plots were situated about 50 m apart, conditions were fairly uniform between them, and thaw depths did not differ by more than 2 cm. In contrast, the GR sites showed a higher average thaw depth increase (0.91 cm per day). Differences in measurements between plots reached up to 11 cm, even though sites were only separated by about 3 m, and vegetation and soil conditions seemed similar.

## 3.2 Carbon Fluxes

All interpolation models described in Section 2.4 yielded a significant linear regression fit between observed and calculated values (for details, see Tab. 2).



### 3.2.1 $CO_2$ fluxes

As reflected in the strong enhancement in both component fluxes of $NEE$, i.e. $GPP$ and $R_{eco}$, the carbon turnover rates in the grazed ecosystem were increased as a response to the warmer and drier conditions in the top soil layers (see Fig. A1).

Regarding photosynthetic uptake of $CO_2$, the average $GPP$ was significantly higher at GR compared to UGR ($p < 0.0001$, see Fig.2). While all three GR plots show higher $GPP$ compared to flux rates at UGR, the average difference between the sites is dominated by differences between the greater fluxes at plot GR-2 and lower fluxes at UGR-2, while differences between GR-1, GR-3 and UGR-1 were not significant.

Across the entire measurement period, ecosystem respiration $R_{eco}$ was distinctly higher at GR compared to UGR. In this

case, site differences were more consistent, i.e. differences in flux rates between plots for each site were minor. Greater $T_{soil}$ and decreasing $SM$ were identified as the main controls for the higher $R_{eco}$ at the GR sites, except for GR-3 where $R_{eco}$ did not increase in response to drying. Possible reasons for that are discussed in chapter 4.2.

For $NEE$, the observed differences in both $GPP$ and $R_{eco}$ canceled out, resulting in no significant changes in $NEE$ as a function of grazing disturbance. Temporal dynamics in $NEE$ largely matched across sites, including a decrease in net uptake

rates during the first and warmer week of the observation period, and an increased uptake during the subsequent, cooler days. GR-2 was found to be the strongest carbon sink in the first period, while $NEE$ at UGR-1 was largest during the second. Overall, all sites were consistent sinks for atmospheric $CO_2$ during the observation period.

### 3.2.2 $CH_4$ fluxes

We observed strong variations in $CH_4$ fluxes between the plots at GR and UGR sites. While flux rates at both sites were

similarly large in the beginning of the experiment, average $CH_4$ fluxes at GR plots started to decline within the first week of observation, in close correlation with decreasing soil moisture. In contrast, the high water table at UGR facilitated high $CH_4$ fluxes throughout the observation period, while changes in time were mostly connected to changes in soil temperatures.

At both sites, the variability in $CH_4$ fluxes across plots for each site was larger compared to the $CO_2$ fluxes. Particularly for GR, flux estimates even between closely co-located plots changed from virtually zero to rates similar to those found at

the ungrazed site. Fluxes at GR-3 were smallest throughout the observation period. During the first days of the experiment, characterized by high soil moisture, $CH_4$ emissions at GR-1 and UGR-2 were largest, while in the second period, when soils at GR dried, they were largest at UGR-2 and UGR-1.

## 4 Discussion

### 4.1 Assessing the Quality of Flux Chamber Measurements

The application of flux chambers may lead to biases in the ecosystem fluxes themselves (Kutzbach et al., 2007). Installing collars in the ground before starting the experiment, while necessary to prevent leaking of air, can disturb the soil and plant roots. At Pleistocene Park, we had to cut a shallow slit in the ground to be able to tightly fit in collars. In a long term study

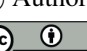



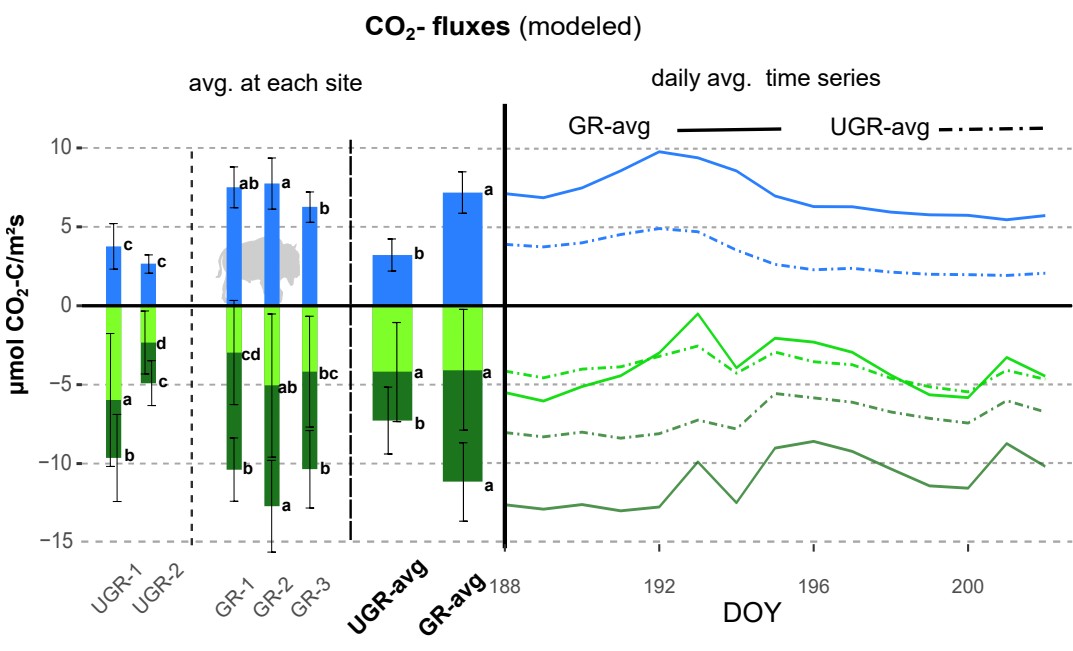

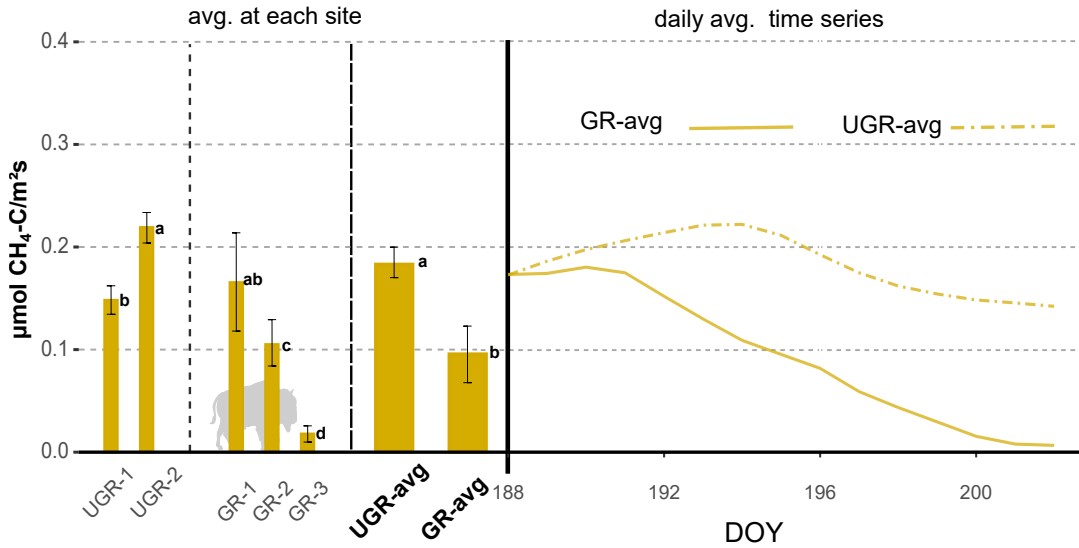

**Figure 2.** Overview on C-fluxes at all chamber plots from July 7th to July 21st. Differing letters indicate significant differences between plots ($p < 0.01$). Overall (site average), $NEE$ did not differ significantly between GR and UGR, while $GPP$ and $R_{eco}$ were significantly larger at GR. Daily average $CH_4$ fluxes at grazed plots strongly decrease over time, leading to a substantial net reduction in methane emissions at the GR plots, compared to the UGR reference.





**Table 2. $R^2$ and $p$-values for linear regressions between final modeled fluxes and measured fluxes.**

|        | UGR-1    | UGR-2   | GR-1     | GR-2     | GR-3     |
|--------|----------|---------|----------|----------|----------|
| $GPP$  | 0.82**** | 0.11*   | 0.87**** | 0.73**** | 0.81**** |
| $R_{eco}$ | 0.90**** | 0.88**** | 0.86**** | 0.44**** | 0.85**** |
| $NEE$  | 0.84**** | 0.49**** | 0.76**** | 0.61**** | 0.80**** |
| $CH_4$ | 0.84**** | 0.93**** | 0.84**** | 0.93**** | 0.88**** |

$****, ***, **, *, ns.$ indicate $p < 0.0001, p < 0.001, p < 0.01, p < 0.05, p > 0.05$, respectively.

**Table 3. Mean daily C-Fluxes for each chamber site (total SE in brackets). Values in $\mu mol C * m^{-2} * s^{-1}$**

| Plot    | $GPP$          | $R_{eco}$    | $NEE$          | $CH_4$          |
|---------|----------------|--------------|----------------|-----------------|
| UGR-1   | -9.55($\pm$2.80)  | 3.67($\pm$1.46) | -5.88($\pm$4.26)  | 0.15($\pm$0.014)   |
| UGR-2   | -4.82($\pm$1.44)  | 2.58($\pm$0.58) | -2.24($\pm$2.02)  | 0.22($\pm$0.015)   |
| **UGR-avg** | **-7.19**($\pm$2.12) | **3.13**($\pm$1.02) | **-4.06**($\pm$3.14) | **0.18**($\pm$0.015) |
| GR-1    | -10.3($\pm$2.03)  | 7.42($\pm$1.31) | -2.88($\pm$3.34)  | 0.17($\pm$0.049)   |
| GR-2    | -12.6($\pm$2.94)  | 7.66($\pm$1.64) | -4.95($\pm$4.58)  | 0.10($\pm$0.023)   |
| GR-3    | -10.26($\pm$2.48) | 6.17($\pm$0.97) | -4.08($\pm$3.55)  | 0.018($\pm$0.008)  |
| **GR-avg** | **-11.06**($\pm$2.48) | **7.09**($\pm$1.31) | **-3.97**($\pm$3.82) | **0.1**($\pm$0.028) |

assessing both chamber and eddy-covariance fluxes conducted in the Alaskan tundra, disturbing roots and soil during chamber

setup was shown to have a depressing effect on both $R_{eco}$ and, to a stronger extent, $GPP$. The chamber fluxes only caught up

with the eddy-covariance fluxes a few years after installing the chambers, most likely linked to a regeneration of belowground

structures after the initial disturbance (Celis et al., 2017). In case of our study, the collars at UGR had already been installed

six years before the start of the experiment, and accordingly the disturbance effect should be negligible. In contrast, installation

artifacts are likely at GR, potentially leading to underestimated fluxes, mainly considering $GPP$. The observed enhancement

in both $GPP$ and $R_{eco}$ following drainage may therefore be a conservative finding. A net effect on the derived shifts in $NEE$

is also possible, since $GPP$ and $R_{eco}$ can be affected in different ways, but a quantification of the potential bias cannot be

done without a longer-term observational dataset.

Another important aspect concerning representativeness of chamber-based carbon flux measurements is small-scale hetero-

geneity in the ecosystem, which may exist even within plots that are seemingly homogeneous. This heterogeneity is observable

at sub-meter scales, and can be a result of disturbances by soil fauna, pockets of fine root proliferation, moisture gradients, or

remnants of decaying organic matter (Davidson et al., 2002). In the Arctic tundra, this small-scale heterogeneity is common

(Aalto et al., 2013; Zona et al., 2011), and is e.g. reflected by variations in soil temperature, soil moisture and thaw depths. To

account for it in flux uncertainties, more than one chamber is needed to adequately assess the mean and variance of surface-



atmosphere exchange fluxes (Davidson et al., 2002). At UGR, spatial variability of carbon fluxes and environmental conditions was analyzed in detail along a transect of ten flux chamber plots in a previous study (Kwon et al., 2016), demonstrating that the two positions chosen for the presented study (IDs 2-0, 2-2) feature fluxes very close to the mean flux rates across all plots, and can therefore be assumed to be representative for the larger area. At GR, the relatively low number of plots available for this study, combined with the lack of previous studies within the area, implies that a larger scale representativeness of this part of the dataset cannot be validated. In combination with the short temporal coverage, the GR dataset should therefore be regarded as a snapshot in both space and time, demonstrating that there is the potential for significant changes in carbon cycle processes following grazing disturbance in permafrost wetlands, while not necessarily providing representative flux quantification for a larger domain. As a guideline to evaluate the captured variability across plots, coefficients of variance (CV) can be computed. CVs of $R_{eco}$ measurements in seemingly homogeneous ecosystems typically range around 30%, while reaching higher values for $CH_4$ fluxes, which tend to be more location-specific (Davidson et al., 2002). For our study, CV for $R_{eco}$ at GR was 11%, but reached 20% at UGR. For $CH_4$, CVs were 77% and 32%, respectively. The characteristics found in our dataset are therefore generally in line with findings presented by Davidson et al. (2002), with the low CVs in $R_{eco}$ potentially linked to the relatively low number of plots observed.

### 4.2 Environmental parameters controlling carbon fluxes: Implications for interpolation

Generally, $R_{eco}$ in wet tundra and peatlands is enhanced by warmer temperatures due to increased microbial activity (Ueyama et al., 2014; Aurela et al., 2007; Kwon et al., 2016), and it may also increase due to drying, with increased potential for aerobic respiration (Lafleur, 2009; Kittler et al., 2017; Kwon et al., 2016). Identifying and comparing the controls for $R_{eco}$ resulted in different equations across the GR and UGR sites, linked to the fact that their vegetation and soil structures differed, as described in detail below.

The GR site is a flat meadow with dense grass tillers, some decaying tussocks, and relatively dry and warm soils. At UGR, soils are water-logged, interspersed with tussocks raised above the water level, and the relatively cool, wet soils are covered by a thick organic layer. These differences justify the use of different sets of controls to explain temporal variability in $R_{eco}$. At UGR, we used only $T_{air}$ as a driver for interpolating $R_{eco}$, since the water table was constantly above ground, and accordingly fluctuations in soil moisture were ruled out. At GR, we used $T_{soil}$ in 5cm and $SM$ in 7.5cm instead to explain variability in $R_{eco}$. Our choice is in line with other studies correlating $T_{soil}$ and $SM$ with $R_{eco}$ (Huemmrich et al., 2010). Hereby it is important to remind, that we had only one set of instruments to measure $T_{soil}$ and $SM$ at GR. However, thaw depths and $CH_4$-fluxes (an indicator for wet/anaerobic conditions (Kwon et al., 2016)) at the grazed plots varied strongly at this small scale. Therefore, we concluded that the measured soil conditions are not fully representative across the three sub-plots. For that reason, at GR-3 also $T_{air}$ was used to interpolate $R_{eco}$, since the coefficient of determination was much higher compared to $T_{soil}$. No statistical relationship between $R_{eco}$ and $SM$ could be found for this plot, probably also because $SM$ actually behaved quite differently compared to the actual measured values.

Also $CH_4$ fluxes varied strongly between GR plots, being highest at GR-1 and lowest at GR-3. Principally, high $T_{soil}$ and water saturated conditions promote a high $CH_4$ release (Kwon et al., 2016). We observed an almost continuous cooling trend



from warm to cold across the whole observation period. At the same time, also soil moisture continuously decreased. Therefore, high $SM$ were always accompanied by high $T_{soil}$ and low $SM$ by low $T_{soil}$. This made it difficult to disentangle their effects on $CH_4$ fluxes.

### 4.3 Chamber fluxes in Arctic tundra ecosystems

Estimates of carbon fluxes obtained by chamber measurements in this study cover a similar range compared to recent reference studies related to grazing in the arctic tundra (see Tab. 4). Still, when directly comparing the results between studies, one has to keep in mind inter- and intra-annual variability in environmental conditions, and therefore also in carbon fluxes, which are highly pronounced in tundra ecosystems (López-Blanco et al., 2017; Falk et al., 2015). Differences in experimental approaches how to assess and display the fluxes further aggravate the comparison of fluxes across studies. Furthermore, many studies cover longer timescales compared to this study.

Values presented by Kwon et al. (2016), obtained during flux chamber campaigns from the period 2013-2015 at the Ambolikha site that served as the UGR reference within the context of the presented study, largely agree with our findings. A wet *Carex* meadow (grazed by geese) in a subarctic coastal tundra showed very similar values compared to values at UGR (Kelsey et al., 2016). Most of the other studies displayed in Tab.3 were conducted in the high arctic, therefore flux magnitudes are expected to be lower compared to our study (Cassidy et al., 2016; Falk et al., 2015; Curasi et al., 2016). Falk et al. (2015) observed similar patterns concerning the amplification of both $GPP$ and $R_{eco}$ as a response to grazing compared to our study for a wet tundra ecosystem. However, we are not aware of studies examining carbon fluxes on a similar type of a grazed ecosystem, compared to Pleistocene park, in corresponding climatic conditions.

**Table 4. Comparison of mean growing season fluxes (chamber measurements) in recent studies. Values in $[\mu mol C m^{-2} s^{-1}]$. (\*) indicates sites in the same area as UGR in this study.**

| Year | $GPP$ | $R_{eco}$ | $NEE$ | ecotype | Reference |
|------|-------|-----------|-------|---------|-----------|
| 2019 | -11.06(±2.48) | 7.09(±1.31) | -3.97(±0.59) | wet tussock tundra (GR, grazed) | this study |
| 2019 | -7.19(±2.12) | 3.13(±1.02) | -4.06(±3.14) | wet tussock tundra (UGR, ungrazed) | this study |
| 2014 | -7.32 (±0.11) | 3.15(±0.15) | -4.15 (±0.17) | wet tussock tundra(*) | Kwon et al. (2016) |
| 2014 | -5.98 (±0.03) | 3.84(±0.20) | -2.14 (±0.17) | wet tussock tundra(*) | Kwon et al. (2016) |
| 2016 | -7.17 (±0.33) | 5.54(±?) | -1.63 (±0.33) | grazed wet *Carex* meadow | Kelsey et al. (2016) |
| 2016 | -4.26 (±0.61) | 2.69(±0.26) | -1.60 (±0.56) | high arctic tundra | Curasi et al. (2016) |
| 2012 | -4.67 (±0.32) | 1.91(±0.1) | -2.73 (±0.26) | arctic mire, grazed | Falk et al. (2015) |
| 2012 | -4.28 (±0.34) | 1.67(±0.076) | -2.53 (±0.26) | arctic mire gr. exclosure | Falk et al. (2015) |
| 2013 | -3.91 (±0.21) | 2.43(±0.1) | -1.47 (±0.15) | arctic mire, grazed | Falk et al. (2015) |
| 2013 | -3.07 (±0.21) | 2.29(±0.11) | -0.78 (±0.16) | arctic mire, gr. exclosure | Falk et al. (2015) |
| 2015 | -1.47 (±0.26) | 1.14(±0.15) | -0.33 (±0.15) | ungrazed high arctic tundra | Cassidy et al. (2016) |





**4.4 Grazing Impacts on Vegetation and Albedo at Pleistocene Park**

Grazing by large herbivores had a number of obvious impacts on the vegetation in Pleistocene Park. However, one issue that complicates the attribution of the herbivore influence on the vegetation is the year-long human disturbance by the park operations. While this influence is mostly restricted to selected areas and transects across the park, and no major direct impact was apparent for the study plots used within the context of this study, one cannot completely disentangle the impacts of man-made

and grazing disturbances. Another issue that needs to be considered when interpreting the presented intercomparison of GR and UGR sites are potential differences in site characteristics that were already present before grazing management in the Pleistocene Park areas started. Lacking pre-treatment flux datasets for both sites, the only reference that is available is the similar appearance of both GR and UGR sites in photographs from the early 2000s, i.e. a time when ecosystem transformation due to grazing was still in an initial stage. Accordingly, while the differences in aboveground ecosystem characteristics described

below can clearly be attributed to grazing pressure over the past two decades, the highlighted differences in carbon fluxes may at least partially have been present in the pre-treatment state. While unlikely to affect the qualitative tendencies of higher carbon turnover, and reduced methane emissions, this potential systematic bias clearly needs to be considered when evaluating and interpreting the flux numbers.

Around our chamber site at GR, almost all sedge-tussocks were in a state of decay, or had disappeared almost completely.

In place of them or between their remnants, many single plant tillers (mainly *Carex spec.* and *Calamagrostis langsdorfii*) grew. These apparent changes in soil and vegetation properties in Pleistocene park are in accordance with previously reported observations documenting grazing impacts. For example, the transformation from tussocks to grass mats by grazing, accompanied by a strong increase in belowground biomass, was already observed for montane biomes (Hofstede and Rossenaar, 1995; Pucheta et al., 2004). Some sedges found in Arctic environments, such as *Carex aquatilis*, were shown to benefit from muskox-grazing,

since they feature strong root production and the ability to produce dense grass tillers, and therefore more easily recover from grazing (Raillard and Svoboda, 1999; Kitti et al., 2009). This ability gives them an advantage over other species (Tolvanen and Henry, 2000), leading to a more turf-like vegetation structure that gradually replaces the original plant community.

Fertilization of tundra ecosystems through available nutrients from urine and faeces also influences vegetation communities under grazing pressure (Raillard and Svoboda, 1999, 2000). Accelerated urea-nutrient uptake by living plants has been reported

for upland tundra (Barthelemy et al., 2018), where graminoids were more efficient in using these resources compared to shrubs. At ungrazed sites such as UGR, the aboveground parts of the plant die off, wither and accumulate on the topsoil, where they rot slowly, leading to a thick organic layer (Kwon et al., 2016). In comparison, at grazed sites such as GR, the plant shoots were regularly removed by the animals resulting in reduced litter accumulation. Linked to the same effect, valleys in the Canadian Arctic which are regularly grazed by muskoxen give the impression of a productive meadow, while ungrazed sites in the same

region appear overgrown, and rather nutrient starved (Raillard and Svoboda, 2000). Similar effects were observed by Falk et al. (2015), where excluding muskoxen from an Arctic mire decreased the amount of plant tillers and increased litter and moss cover. In both upland and lowland tundra ecosystems, herbivores, mostly reindeer or muskoxen, have been shown to reduce moss cover, and decrease shrub cover by trampling and browsing, promoting the expansion of graminoids (Kitti et al., 2009;





Manseau et al., 1996; Olofsson, 2006; Ylänne et al., 2018; Falk et al., 2015). Landscapes covered by graminoids usually have

a higher albedo compared to shrub covered ones (Te Beest et al., 2016; Chapin et al., 2005). Accordingly, herbivore grazing can systematically increase the surface reflectivity, and therefore reduce ecosystem energy input. This observation agrees with our results, which show a significantly higher albedo at GR compared to UGR. Within Pleistocene Park, we are confident that in more shrubby, heavily browsed upland tundra and taiga areas, the increase of albedo following the grazing impact is even more pronounced due to the expansion of graminoids. Since the grazing history of about 22 years at Pleistocene park is still

relatively short, the ecosystem is most probably still in a transition state. We therefore expect further changes in vegetation community structure over the coming decades, given a persistent grazing pressure, i.e. a further condensing of the grass mat and accordingly increased tiller formation and living belowground biomass. Denser grass cover is likely to further enhance the albedo, and also will more effectively shade the soil surface. On the long term, both effects should therefore contribute to alter the energy budget, mostly likely leading to a cooling of shallow soil layers (see also below).

**4.5 Grazing impacts on soil properties**

We found overall $T_{soil}$, especially in the topsoil, to be higher at the grazed sites, as well as temperature gradients to be steeper, and thaw depths to be greater. At the same time, GR was relatively dry compared to UGR, where the water table was above ground throughout our observation period. Ultimately, $T_{S,5cm}$ at GR reacted one order of magnitude faster to changes in $T_{air}$ compared to UGR-1 and UGR-2. Overall, we suggest two dominating processes how grazing pressure transformed the soil

at GR: Compacting the soil by trampling the organic peat layer (plus related effects on vegetation) and the reduction of soil moisture. A long-term drainage experiment conducted at UGR demonstrated that the topsoil peat layer can strongly influence the soil thermal regime within tundra ecosystems. When drying out the organic layer, it warms up faster, but conducts the heat to deeper soil layers only very inefficiently (Kwon et al., 2016). When this peat layer is trampled by herbivores, as observed at GR, the soil thermal regime is significantly modified. Furthermore, it probably leads to fewer air-filled pores in the overall

soil profile and creates higher soil bulk densities, and in turn also increases thermal conductivity and diffusivity (Ochsner et al., 2001).

A reduction of soil moisture, which was observed at GR, generally leads to a decrease of thermal conductivity and heat capacity in organic soils (Ochsner et al., 2001; Kwon et al., 2016). Since the warming of the soil was overall stronger at GR, the effects of lower water content on reducing the soil heat capacity along with the higher conductivity caused by topsoil

compaction may have taken over as the dominant control. Accordingly, higher temperatures and temperature fluctuations are likely, since less energy is needed to warm up the soil and increase thaw depths. In previous studies, trampling and grazing in the tundra was already shown to lead to the diminishing of the organic litter/moss layer on the topsoil, the decrease of shading by shrubs, and consequently causes higher soil temperatures and more compact soils (Ylänne et al., 2018; Olofsson et al., 2001, 2004; Te Beest et al., 2016; Falk et al., 2015). These studies suggest that the differences in soil properties between GR

and UGR may be predominantly attributed to grazing pressure.

As stated in the previous section, belowground biomass is expected to increase as the grass mat at GR gets more and more dense. A higher percentage of organic matter in the soil would decrease heat conduction (Balland and Arp, 2005; Abu-Hamdeh





and Reeder, 2000), and therefore it can be expected that heat conductivity will decrease again, as the transformation of the ecosystem progresses. Accordingly, on the long term thaw depths may become shallower, and carbon pools in deeper layers may be better preserved as temperatures drop.

The lower soil moisture observed at GR can be linked to an increase in evapotranspiration. The transpiration rate correlates with an observed stronger photosynthetic activity (i.e. $GPP$), due to the adjustment of stomatal conductance to match the biochemical potential for photosynthesis (Farquhar and Sharkey, 1982; Field et al., 1992). Additionally, evaporation increases due to the decrease of litter as a result of grazing (Larson and Whitman, 1942; Dyksterhuis and Schmutz, 1947). Accordingly, a decline in soil moisture is reported for tundra ecosystems under grazing influence (Ylänne et al., 2018), and also for other grazed biomes (Vandandorj et al., 2017; Yan et al., 2018; Larson and Whitman, 1942; Dyksterhuis and Schmutz, 1947), where soil structural (i.e. compaction) and subsequent hydrological changes have been reported. In summary, our results for the grazed site within Pleistocene Park, which show drier soils and a higher $GPP$ compared to UGR, agree with previous findings, emphasizing that grazing can exert strong changes to the thermal and hydrological regimes.

## 4.6 Grazing influence on carbon fluxes

Our findings confirm that grazing in tundra ecosystems can lead to higher $T_{soil}$ and lower soil moisture, which is usually related to an increase in $R_{eco}$ (Ylänne and Stark, 2019; Väisänen et al., 2014; Metcalfe and Olofsson, 2015; Cahoon et al., 2012). This increase in $R_{eco}$ was similar in experiments studying warming and drainage effects on tundra ecosystems exclusively (Christensen et al., 2000; Huemmrich et al., 2010; McEwing et al., 2015; Oechel et al., 1998; Zona et al., 2011; Natali et al., 2015). In addition to triggering these biogeophysical shifts in soil properties, grazing was also shown to lead to faster nutrient cycling in both upland and lowland tundra ecosystems (Ylänne et al., 2018; Olofsson et al., 2001, 2004; Stark et al., 2007; Barthelemy et al., 2018; Raillard and Svoboda, 2000). Increases in primary productivity linked to higher nutrient availability (Olofsson et al., 2001; Ylänne and Stark, 2019; Raillard and Svoboda, 2000), along with soil thermal and hydrological changes, might therefore be an explanation for high $GPP$ fluxes observed in our Pleistocene Park study plots. As discussed earlier, these increases may even be more pronounced after full recovery from the disturbance inflicted by installing the chamber collars.

Accordingly, despite the reported increases in $GPP$, our experiments did not yield an enhancement in net carbon uptake by grazing, i.e. a more negative $NEE$, during the growing season. In contrast, in other wet Arctic graminoid communities (Falk et al., 2015), NEE was greatly enhanced by grazing. In upland tundra ecosystems, however, net uptake was often lower at grazed compared to ungrazed or only lightly grazed sites. Main explanations for these shifts were either a simultaneous increase in $R_{eco}$ at grazed sites (Ylänne and Stark, 2019; Väisänen et al., 2014; Metcalfe and Olofsson, 2015; Cahoon et al., 2012) balancing $GPP$ gains, or a decrease in $GPP$ linked to a decrease in plant biomass and leaf area index (Metcalfe and Olofsson, 2015; Cahoon et al., 2012) following grazing. However, at the same time grazed upland tundra exhibits stable or even increased below-ground carbon storage (Ylänne et al., 2018), indicating that decreased $R_{eco}$ in the non-growing season possibly (over-)compensates the decreased $NEE$ in the growing season. It can be speculated that the reported differences in ecosystem properties between grazed and not grazed areas have further implications for carbon fluxes in the non growing season that have not yet been studied in detail.





GR is an ecosystem in transition from a tussock tundra topped by a thick organic layer to a grassland with dense tillers. At the same time, the mean water table during the growing season has been lowered. These shifts also change in which soil horizon carbon is primarily accumulated, with stronger accumulation in deeper soil layers, as observed by Hofstede and Rossenaar (1995) and Pucheta et al. (2004). The decay of the remnants of tussocks, as well as the decomposition of large carbon pools in the now drier organic topsoil, contribute to the observed high ecosystem respiration rates. These fluxes, however, will probably decrease in the future, leading to decreasing rates of $R_{eco}$ and therefore increased $NEE$.

In contrast to the current situation in most circum-Arctic ecosystems, Pleistocene Park exhibits a high diversity of herbivores that are absent from the landscape elsewhere. Previous studies indicated that herbivore diversity leads to a more balanced use of food plants (Chang et al., 2018, 2020; Larter and Nagy, 2001; Sitters et al., 2020; Cromsigt et al., 2018) and correlates with increased carbon uptake and soil carbon storage (Chang et al., 2018, 2020; Sitters et al., 2020). While such shifts yet have to be shown for permafrost ecosystems, first observations in Pleistocene Park hint at positive long-term effects of big herbivore introduction on carbon sequestration. These insights, and also questions emerging from ongoing studies, call for future research concerning the influence of big herbivores on the carbon balance of permafrost ecosystems, with a focus on long-term, year-round monitoring.

## 5 Conclusions

In this study, we investigated the impact of long-term grazing disturbance on a previously wet tussock tundra ecosystem underlain by permafrost in the Siberian Arctic using flux-chamber observations over 2.5 weeks during the growing season in summer 2019. Over the past 22 years, introduction of large herds of herbivores in the context of the so-called Pleistocene Park experiment has altered vegetation and soil properties within the affected area, this way initiating an ongoing transformation from a water-logged, overgrown tussock tundra towards a drier ecosystem featuring more turf-like vegetation. We compared the managed ecosystem inside Pleistocene Park to a nearby undisturbed reference site, focusing our study on differences in soil thermal and hydrological properties, and how these influenced the exchange fluxes of carbon between ecosystem and atmosphere.

We measured a significantly lower albedo at the grazed site compared to the undisturbed reference, which can be mostly explained by a lower abundance of shrubs. Soil compaction as a result of trampling, in combination with higher evapotranspiration losses, led to a decrease in soil moisture. Linked to the associated reduction in soil heat capacity, topsoil temperatures in the park were higher and reacted one order of magnitude faster to changes of air temperatures compared to the undisturbed tundra. Due to warmer and drier conditions in the soil, both $GPP$ and $R_{eco}$ during July were significantly higher at the grazed site in the park compared to a undisturbed wet tussock tundra, while differences in $NEE$ are not pronounced. $CH_4$ emissions, following the shift in hydrological properties, were distinctly lower in the park, but also highly variable between plots.

The effect of grazing on nutrient availability, and associated responses of the vegetation community, remain open questions that must be quantitatively assessed at Pleistocene park. Furthermore, it is essential that carbon fluxes will be investigated over longer timescales, with year-round data coverage. Especially fluxes during autumn and early winter, which account for a





significant part of the annual carbon budget, need to be included to enable a more comprehensive assessment of the net effects of grazing management on carbon sequestration in the Arctic tundra. Accordingly, future experiments are planned to address these research topics.

*Code and data availability.*  Both datasets and code are available from the author upon request.





## Appendix A: Derivation of Models for the Interpolation of C-Fluxes

### A1 Modeling $R_{eco}$

Investigating the environmental drivers and their evolution in correlation with $R_{eco}$ measurements revealed that there is no uniform set of drivers across observation sites yielding optimum regression fits across.

For GR-1 and GR-2, in contrast to all other sites, changes in $SM$ apparently exerted a strong influence on $R_{eco}$ (see Fig.A1). However not in all $SM$-ranges both high and low $T_{soil}$ or $T_{air}$ were measured. Therefore, $R_{eco}$ fluxes during a moisture interval
that shows a wide spectrum of soil temperatures, specific for each site ($SM_{7.5cm}$ at GR-1: 54% - 61%; at GR-2: 57.5% - 62%), were chosen in which a exponential regression ($T_{soil}$ in 5 cm $\sim R_{eco}$) was conducted (blue dots, left graphs, Fig. A1). The resulting formula was applied to the $T_{soil}$ in 5 cm of the whole SM range, which yielded a simulated set of values for $R_{eco}$ if soil moisture did not change. This set of values was subtracted from the original flux values, and residuals were utilized for a second, linear regression (residuals $\sim SM_{7.5cm}$, central graphs, Fig. A1) to account for the influence of variable soil
moisture on fluxes. Huemmrich et al. (2010) observed a similar correlation between soil water regime, soil temperature and soil moisture as proposed here, substantiating the approach. For $R_{eco}$ estimates at GR-3, UGR-1 and UGR-2, fits were optimal when utilizing air temperatures in combination with an exponential regression, while no significant correlation could be found when trying to explain residuals with $SM$, thaw depth or other variables (Fig A2).





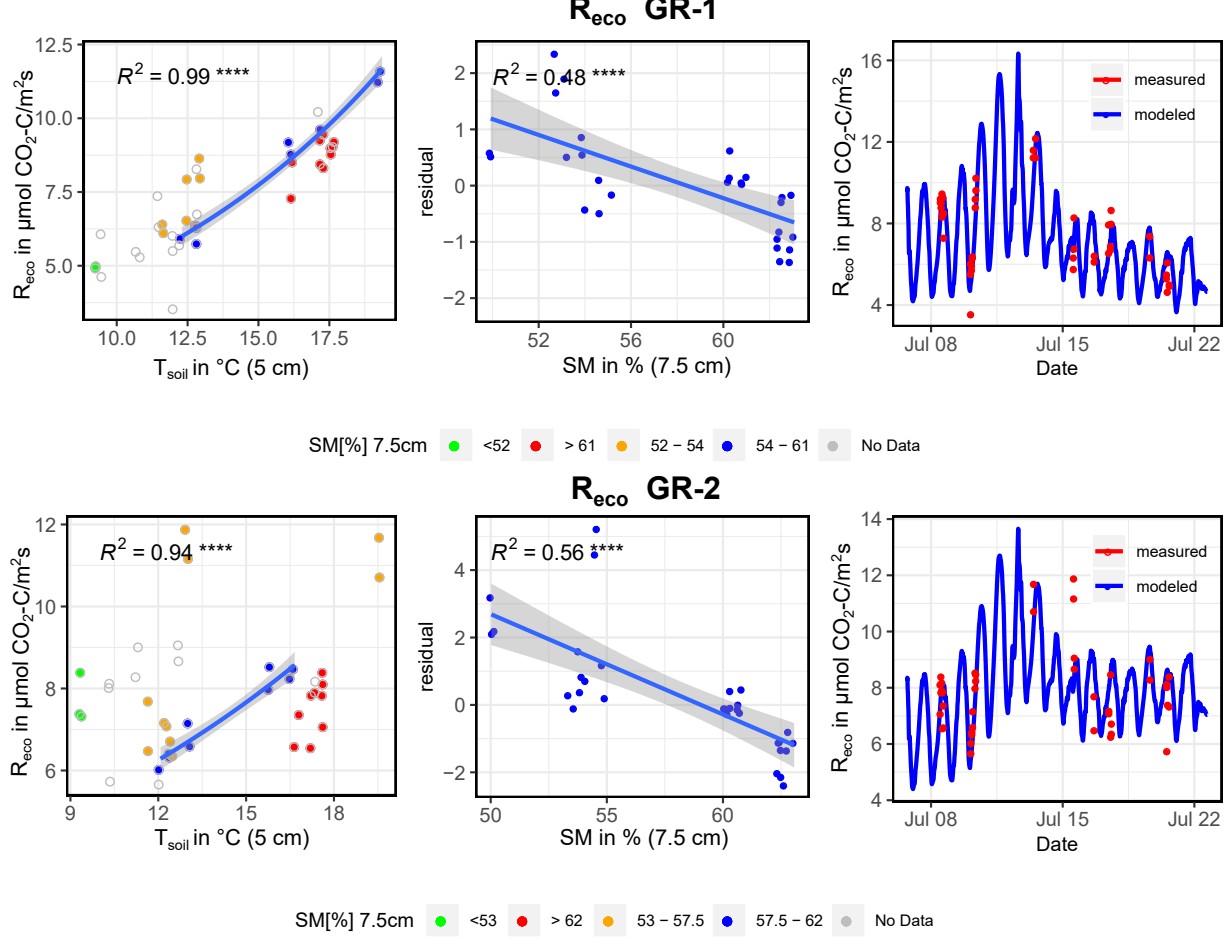

**Figure A1.** Depiction of the relationship between $T_{S,5cm}$ and $SM$ and $R_{eco}$ for GR-1 and GR-2. Interpolation models are formed by the equations of depicted regression curves. The graphs on the right show modeled vs. measured fluxes, respectively.



**Figure A2.** Depiction of the relationship between $T_{air}$ and $R_{eco}$ for GR-3, UGR-1 and UGR-2 (left column). Interpolation models are formed by the equation of the depicted regression curve. The graphs on the right show modeled vs. measured fluxes, respectively.






## A2 Modeling $CH_4$ Fluxes

$CH_4$ fluxes showed a strong correlation to both $T_S$ (all sites) and $SM$ at all depths (GR-1, GR-2, GR-3). However, there was a strong co-linearity between $T_S$ and $SM$. Therefore, to reach the best possible fit for interpolating $CH_4$ fluxes, at GR, while accounting for both drivers, data was split up in two moisture groups ($SM_{15cm} > 60\%$ and $SM_{15cm} < 60\%$) to apply

a pseudo-stepwise regression. Then, for each plot, a linear regression between $CH_4$ fluxes of the lower moisture group and $T_{S,25cm}$ was computed (Fig.A4 $b, e, h$). Second, the resulting linear equation was applied to the complete dataset for each plot integrating both moisture groups. The residuals between these calculated values and the measured values was fitted against $SM_{15cm}$, applying another linear regression, resulting in a second linear equation. These two resulting equations were used to interpolate $CH_4$ fluxes for each plot. Since soil moisture did not change at UGR-1 and UGR-2, the linear regression between

$CH_4$ fluxes and $T_{S,15cm}$ yielded the linear equations used to model fluxes at these sites (Fig.A3).

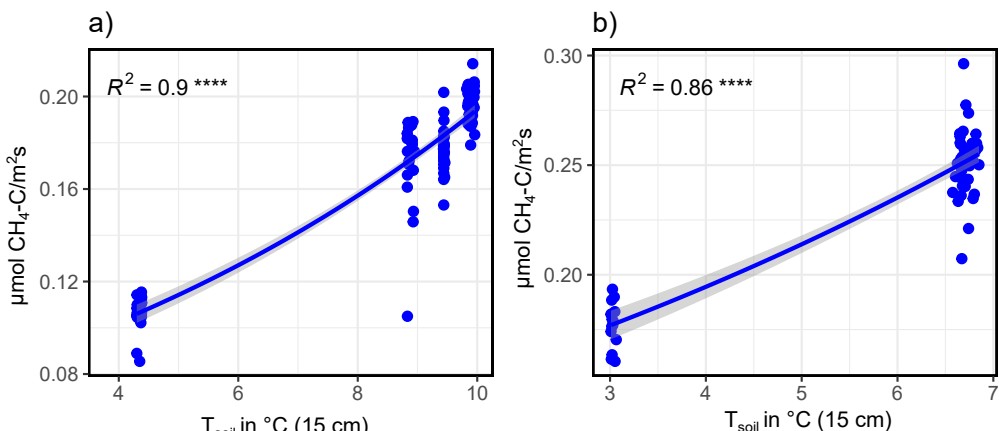

**Figure A3.** Exponential regressions between $CH_4$ fluxes and $T_{S,15cm}$ for UGR-1 ($a$) and UGR-2 ($b$).

**Figure A4.** Depiction of influencing drivers of $CH_4$-fluxes (GR-1: $a, b, c$, GR-2: $d, e, f$, GR-3: $g, h, i$) and the following derivation of formulas for the interpolation process, showing how magnitude of fluxes is higher for high soil moisture $(a, d, g)$, and how $SM_{15cm}$ and $T_{S,25cm}$ jointly explain $CH_4$ fluxes.





## A3 Error Calculation

**Table A1. Error range of C-fluxes, with values given in $[\mu mol C m^{-2} s^{-1}]$. $Err_{abs}$ describes the final cumulative error that is also used in the results section.**

|  | UGR-1 | UGR-2 | GR-1 | GR-2 | GR-3 |
|---|---|---|---|---|---|
| **$R_{eco}$** |  |  |  |  |  |
| $Err_{slope}$ | 0.16 | 0.13 | 0.15 | 0.13 | 0.13 |
| $Err_{mod}$ | 0.89 | 0.45 | 0.65 | 1.16 | 0.84 |
| $Err_{T_S}$ | - | - | 0.51 | 0.35 | - |
| $Err_{abs}$ | 1.46 | 0.58 | 1.31 | 1.64 | 0.97 |
| **GPP** |  |  |  |  |  |
| $Err_{slope}$ | 0.30 | 0.24 | 0.26 | 0.27 | 0.33 |
| $Err_{mod}$ | 2.50 | 1.20 | 1.77 | 2.67 | 2.15 |
| $Err_{abs}$ | 2.80 | 1.44 | 2.03 | 2.94 | 2.48 |
| **NEE** |  |  |  |  |  |
| $Err_{abs}$ ($R_{eco}$) | 1.46 | 0.58 | 1.31 | 1.64 | 0.97 |
| $Err_{abs}$ (GPP) | 2.80 | 1.44 | 2.03 | 2.94 | 2.48 |
| $Err_{comp}$ | 4.26 | 2.02 | 3.34 | 4.58 | 3.55 |
| **$CH_4$** |  |  |  |  |  |
| $Err_{slope}$ | 0.0022 | 0.0033 | 0.0039 | 0.0026 | 0.00090 |
| $Err_{mod}$ | 0.011 | 0.012 | 0.045 | 0.020 | 0.007 |
| $Err_{T_{S,15cm}}$ | 0.0013 | 0.0014 | - | - | - |
| $Err_{abs}$ | 0.014 | 0.016 | 0.049 | 0.027 | 0.008 |

For the final modeled fluxes, which provide the basis to calculate daily average fluxes, a series of error sources was identified
(see Tab. A1). First, a bootstrapping approach to obtain a median slope of $CO_2$ and $CH_4$ concentration changes (see also methods section) allows to generate an error range for observed flux rates. The standard error of the calculated slopes was transformed into a flux by the same formula applied to the median slope, averaged over all measurements, and is called $Err_{slope}$. For $GPP$, $Err_{slope}$ is composed by both the $Err_{slope}$ of $NEE$ measurements and $R_{eco}$ measurements, since these two direct flux measurements needed to be combined for GPP. Second, modeling the chamber fluxes in order to have
a continuous time series results in deviations from the modeled vs. the measured fluxes. Here, a linear regression (modeled vs. measured) was applied to evaluate the model quality, and to obtain a standard error. Third, to model and interpolate $R_{eco}$ (at GR-1 and GR-2) and $CH_4$ fluxes (at UGR-1 and UGR-2), interpolated soil temperatures were used. Therefore, the RMSE of these models was considered by adding it to the $T_S$ - term in the interpolation formula for $R_{eco}(T_{S,5cm})$ and $CH_4$ fluxes ($T_{S,15cm}$; $T_{S,25cm}$). In a last step, the initial flux was subtracted from this "enhanced" flux, and the result was defined as the





$T_S$-error ($Err_{T_S}$). All these errors are summarized by $Err_{abs}$ and are depicted in Tab. A1. $NEE$ errors are summed up $Err_{abs}$ from $R_{eco}$ and $GPP$.



*Author contributions.* MG was responsible for study conception. WF, MG and CT contributed to development of the methodology. NZ provided logistics at the study site, and has a overall prominent role in designing permafrost studies within Pleistocene Park. WF conducted the field work, with support by MG and NZ. Formal data analysis and interpretation was carried out by WF under supervision of MG and

CT. WF led the drafting of the manuscript with contributions from MG. All authors contributed to refining the manuscript.

*Competing interests.* The authors declare that they have no conflict of interest.

*Acknowledgements.* This work was supported through funding by the European Commission (INTAROS project, H2020-BG-09-2016, Grant Agreement No. 727890, Nunataryuk project, H2020-BG-11-2016/17, Grant Agreement No. 773421), and the German Ministry of Education and Research (KoPf project, Grant No. 03F0764D). Further funding was provided by the Max Planck Institute for Biogeochemistry (MPI-

BGC) in Jena, Germany. The authors appreciate the technical support of Olaf Kolle and Martin Hertel from the Field Experiments and Instrumentation service group at MPI-BGC, and also the contribution of staff members of the Northeast Scientific Station in Chersky for facilitating the field experiments.



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
