# Peer review of "Grazing enhances carbon cycling, but reduces methane emission during peak growing season in the Siberian Pleistocene Park tundra site"

_Biogeosciences, 2021_

## Referee Comment (RC2)

**Review of bg-2021-110**

**Grazing enhances carbon cycling, but reduces methane emission in the Siberian Pleistocene Park tundra site**

**General comments**

The authors provide a nice introduction into herbivory impacts on permafrost ecosystems. The study provides a very interesting insight into ecosystem changes under grazing pressure. The data set used is a measurement series of NEE and $R_{eco}$, measured for two weeks at a grazed and an ungrazed site with several replicates. The observed flux changes in $CO_2$ and $CH_4$ are well described and put into relation with animal activity, such as soil compaction and drying, which shows a significant reduction in $CH_4$ emissions from grazed sites.

The methods used are suitable to use for the provided explanation of these effects, however, the method description itself should provide more detail on the approach.

There are several further topics arising from this study, such as the influence of vegetation species on fluxes and how fluxes change throughout different seasons. It would be great to have more comparison to other studies regarding this.

There is a minor lack of context regarding the general hypotheses of the Pleistocene Park experiment as to why the findings from this study suggest a different effect of animal grazing than previously hypothesized by Zimov et al. (2005). The findings should also be discussed in relation to those hypotheses.

**Specific comments**

Please consider making the data accessible via a scientific data repository.

L89: Please add a map indicating the sampling sites.

L91: There is a new paper by Reinecke et al. (https://doi.org/10.1038/s41598-021-92079-1) dealing with the Pleistocene Park vegetation in more detail, which you should consider here.

L151: Please describe the bootstrapping approach in more detail (number of iterations etc.).

L206: How did you test for significance?

Figure 2: For $CH_4$, it should be clearly stated that these are emissions only. Using "fluxes" suggests a bi- or omnidirectional gas exchange.

Table 2: I assume "ns" means "not significant"? Please make the caption overall more clear. Also, please add something like "ungrazed sites (UGR-1 and -2) and grazed (GR-1, -2 and -3)" to the title of this table. I suggest, for uniformity, to switch axes of this table to make it similar to table 3.

L295: What about previous disturbances of the soil itself, especially in the active layer with freeze-thaw cycles? Please consider this in your manuscript

L357: These pre-existing site differences are very likely, taking the distance between the sites into account. Especially the differences in thaw depth (greater thaw depth at UGR) are opposing the general hypothesis of large animal impact on permafrost ground as a conservation mechanism, which is said

to mainly originate from snow compaction in winter. Maybe you should elaborate or highlight these a little more and discuss why your findings might differ from named hypothesis.

Figure A1: Please provide letters for each graph (e.g. as in figure A3). Also, adding the equation for each regression curve to the corresponding graph would be good.

Figure A2: Please see the comments on figure A1.

Figure A3: Please add the equations for each regression curve.

Figure A4: Please provide headlines for a), d) and g). Also, it should say somewhere in the graph (not only in the caption) that the graphs show $CH_4$ emissions.

**Technical comments**

Please make "C-Fluxes / C-fluxes / C fluxes" consistent throughout the paper. Maybe consider replacing flux considering my earlier comment

L99: Please put *Betula nana* in italics and capitalize, since it's a species name. Also, please change "willow spec." to "*Salix* sp."

L100: Please change "*lugens*" to "*C. lugens*".

L166: R Studio is just the main software. Please provide the used packages.

L170: Suggestion: "…not uniform across plots even at one site…"

L200: There is a leftover "?" in this line. Also, the test should be named "Mann-Whitney-U-test".

Table 2 caption: inconsistency in * and spaces, please adjust

Line 283: please capitalize

**Review criteria:**

*Does the paper address relevant scientific questions within the scope of BG?*
Yes

*Does the paper present novel concepts, ideas, tools, or data?*
Yes

*Are substantial conclusions reached?*
Yes, to some extent

*Are the scientific methods and assumptions valid and clearly outlined?*
Yes, but method description could be more precise

*Are the results sufficient to support the interpretations and conclusions?*
Some interpretations are a little one-directional, but supported by the data

*Is the description of experiments and calculations sufficiently complete and precise to allow their reproduction by fellow scientists (traceability of results)?*
Somewhat, method explanation needs some more detail

*Do the authors give proper credit to related work and clearly indicate their own new/original contribution?*
Yes

*Does the title clearly reflect the contents of the paper?*
Yes

*Does the abstract provide a concise and complete summary?*
Yes

*Is the overall presentation well structured and clear?*
Yes

*Is the language fluent and precise?*
Yes, just very minor things indicated in the technical comments

*Are mathematical formulae, symbols, abbreviations, and units correctly defined and used?*
Yes

*Should any parts of the paper (text, formulae, figures, tables) be clarified, reduced, combined, or eliminated?*
The Methods section should receive more detailed information on the modelling approach. A figure (map) indicating the sites' positions and relations would be nice.

*Are the number and quality of references appropriate?*
Yes

*Is the amount and quality of supplementary material appropriate?*
Yes

---

## Author Response (AR1)

**Author response to interactive comment RC1 submitted on Jul 02, 2021**

In the document below, the reviewer comments have been copied from the original review and are shown in black font, while the author comments have been added in blue.

General comments

The authors describe the effects of a manipulation of grazing herbivores density in a continuous permafrost tundra ecosystem on carbon fluxes during the growing season and some of its predictors. The core data is an extensive set of NEE and Reco measurements obtained over two weeks at peak growing season over three replicates of the high grazing density system and two replicates of the low grazing density system, and is accompanied by meteorological variables. Overall, gross fluxes GPP and Reco were increased in the high grazing density plots, concomitant with an increase air and soil temperature and a decreased soil moisture content, while NEE was largely unaffected. CH4 fluxes were lower in the high grazing density plots but with high variability between plots. The flux measurement dataset is valuable, and my main concerns lie in the choice of an unbalanced design which hampers statistical evaluation of the results, and that too few details are provided to justify the fact that initial conditions were comparable and throughout the Methods section. I would therefore recommend that these aspects be thoroughly improved before publication and try to provide suggestions for such improvements.

As discussed in more detail in the replies to minor comments below, and also in the discussion before the Discussion paper was accepted for publication, there is no conclusive data material available that allows to demonstrate the status of the grazed ecosystem before the start of the Pleistocene Park experiment in the 1990s. We believe that this has been adequately treated in the discussions section of this manuscript; however, to highlight the shortcoming more prominently, we added a statement to the concluding sentences of the abstract:

"Our results indicate that grazing of large herbivores may promote topsoil warming and drying, this way effectively accelerating $CO_2$ turnover while decreasing methane emissions in the summer months of peak ecosystem activity. Since we lack quantitative information on the pre-treatment status of the grazed ecosystem, however, these findings need to be considered qualitative trends for the peak growing season, while absolute differences between treatments are subject to elevated uncertainty. Moreover, our experiment did not include autumn and winter fluxes, and thus no inferences can be made for the annual NEE and $CH_4$ budgets at tundra ecosystems."

The only documentation on pre-treatment status we can offer is the set of photographs that was presented in our previous reply. As discussed also below, we would prefer not to include it as part of the revised manuscript, but if the editor actually recommends it we could add this material as a new part of the Appendix.

We would like to mention again at this point that many studies covering novel, uncharted scientific territory in regard to method and/or location may be associated with a large uncertainty. From all possible forms of scientific inquiry, our abductive method is therefore speculative, but we strove to provide and include all information at our disposal in support of our results and claims. We believe that the information added allows for a detailed discussion of the shortcomings.

I appreciate that, since initial submission, further information has been added regarding the choice of the two replicates in the UGR site, but still think that beyond the choice of these particular plots, the decision of having only two control plots should at least be better motivated and the limitations it implies better discussed. Overall, that does not entirely appease my concerns regarding the unbalanced study design. On the contrary, by choosing two sites that are close to the average within that transect, rather than e.g. at random, the mean value is preserved but the variance is artificially deflated, possibly biasing comparisons. While this may or may not affect Reco dynamics so much, as the differences are marked and variability seems limited, the differences observed in other variables and in particular CH4 fluxes could be artefactual for this reason. In addition, UGR-2 is described as standing out on several aspects, from the lower soil temperature to the higher time lag of soil vs air temperature, to having only half as many flux measurements as the other plots (so that there is, in total, twice as much flux data for GR than UGR plots), which explains in part the poor measured vs modelled fit of GPP for UGR-2.

The decision to work with only two reference sites (UGR) plots at the Ambolikha site was based on practical considerations. In principle, we could have used up to 10 sampling locations which had been established in earlier experiments. However, plots were spaced 25m apart, meaning that the observation system has to be moved between sites when switching locations, as opposed to the GR sites, which were co-located within a narrow radius. Spending time for moving the system implies less time for actual measurements, which is why we wanted to reduce it. Therefore, the choice was made to only sample two sites.

We realized that the description in the submitted version of the manuscript chosen to justify the UGR site selection was somewhat misleading. While fluxes at the two selected were actually indeed close to the mean fluxes across the transect, our choice was rather motivated by the ecosystem structure. While we cannot give more precise information on the GR sites before grazing started, we know from personal communication that the managed area used to be a waterlogged tussock tundra. Out of the 10 plots that were available at the UGR site, six are dominated by cotton grasses (Eriophorum), with few or no tussocks present (see Figure 8 from Kwon et al. (2016), copied below). Two more sites (IDs 4 and 5) were placed on a small ridge, and were therefore significantly drier, and dominated by shrubs. We therefore selected the only two locations, IDs 0 and 2 in the control section, featuring the desired vegetation structure for investigating the effects of grazing. Studies with a different scope may have enabled a random site selection to improve estimates of uncertainty due to site-specific bias.

[Figure]

The respective description will be adjusted in the revised version of the manuscript text: "At UGR, spatial variability of carbon fluxes and environmental conditions was analyzed in detail along a transect of ten flux chamber plots in a previous study (Kwon et al., 2016), including a description of vegetation community structure and hydrologic status at each plot. Based on these findings, we selected two positions for the presented study (IDs 2-0, 2-2) which best represent the vegetation composition of a water-logged tussock tundra that dominated the grazed site before the Pleistocene Park experiment was started in the 1990s."

I do not see a simple way to solve this issue, but perhaps simulating data based on flux measurements from Kwon et al 2016, using the relationship between the fluxes in the current study and those in Kwon et al 2016, could allow to carry out a sensitivity analysis to determine whether the choice of these 2 replicates affected the findings.

As demonstrated in Figure 8 from Kwon et al. (2016), which is copied above, there is some spatial variability in $CO_2$ flux rates along the control transect, so obviously our flux result from the UGR site would have been affected if we had chosen 2 different sites from the set of ten. However, as already outlined above, it was our scope to work with these two sites chosen as references for our study, since they featured the required ecological characteristics, i.e. vegetation community structure and water level status. Only a different scope would have enabled a random site selection since there were more sites to choose from. We agree that revisiting plots across studies is very desirable for intercomparison and to trace ecosystem behavior through time. However, any of the remaining 8 sites would have either been dominated by cotton grasses, or would have featured dry conditions on an elevated ridge. Therefore, a random choice of two

plots out of the 10 available positions would not provide an appropriate approach to find the most suitable references for the scope of the presented study.

Comparing the "grazed" and "ungrazed" treatments is central to this manuscript, but their identity prior to the experiment is unclear from the information currently provided. For instance, as it is now GR sites are described as a wet lowland tundra that gets flooded every year, while UGR sites are a wet tussock tundra floodplain, it is unclear whether the distinction between the two is intended to avoid repetition or to convey a more fundamental difference between the sites. Beyond the more detailed comments below requesting that more detailed data on initial conditions should be presented if available, I would suggest reorganizing the part of the Methods section where the sites are described. In addition to the general geographic and climatic information about the area, it could be easier to follow if the authors would first describe the similarities between the two sites prior to manipulation (e.g. flooding, vegetation, etc.) before delving into what makes them distinct.

The landscape description the reviewer refers to was indeed intended to avoid repetition. In the revised manuscript version, we will emphasize that both sites are/were actually wet tussock tundra ecosystems. We also extended the paragraph that introduces the two selected study sites to emphasize their similarities in terms of the general landscape setting:

"In the context of this study, we compared carbon fluxes and ecosystem characteristics between two measurement sites in the Chersky region. Both sites are wetland ecosystems situated within the Kolyma lowlands region that are waterlogged for the largest part of the growing season, affected by the flooding regime of the Kolyma River that leads to high water levels around snow melt in early summer. Our undisturbed reference ecosystem is a tussock tundra site situated about 15km south of Chersky on the floodplain of the Kolyma River. Due to the very low natural abundance of grazing herbivores in the region, the influence of grazing disturbance on this dataset can be considered negligible. Situated about 15km further south at the margins of the floodplain, the Pleistocene Park site, hosting a variety of herbivores (sheep, yaks, cows, horses, bison, muskoxen, reindeer), was used to study the effects of grazing on permafrost ecosystems."

Specific comments

L30: Might be worth citing the recent review by Mekkonen et al 2021 found here: https://doi.org/10.1088/1748-9326/abf28b

We will add the suggested reference to the list of papers cited here.

L53-56: It's not clear how Pleistocene Park and the measurements presented here address non-growing season carbon cycling, in fact the abstract (L14-15) explicitly states that those are not addressed. I suggest reformulating or removing the reference to nongrowing season fluxes from this section altogether.

We agree that the sentence is misleading, since our own study does not provide new information on off-season fluxes. We will therefore remove this sentence from the introduction.

L58-63: It's perhaps more a question of personal taste, but I find it a bit confusing to present hypotheses that are not tested in the current study (e.g. above-belowground partitioning of GPP, decrease in respiration from colder permafrost in the winter). I would think these wider considerations about the aims of Pleistocene Park itself would fit better in the discussion, or presented differently than a list of hypotheses near the end of the introduction, as that might give the reader the wrong impression that those are the hypotheses addressed in this study.

We agree that not all hypotheses listed here can be tested against the dataset provided by this study, and therefore the list may be confusing the reader. We will therefore rephrase the beginning of the sentence to 'Some of the underlying hypotheses related to the Pleistocene Park project are ..', and omit the former 2nd hypothesis that referred to a decrease in year-round temperatures, and related effects on e.g. respiration.

L83-86: Can you provide more precise information on that site, such as for how long the density of grazing herbivores has been increased, and by how much compared to the "ungrazed" site which arguably hosts large grazing herbivores in lower densities such as the rest of the Arctic, unless the site is too wet and not visited by grazers?

Particularly in the open landscape in close vicinity of Chersky, hunting pressure prevents the establishment of larger herds of wild animals. Accordingly, due to the very low natural abundance of grazing herbivores in the region, the influence of grazing disturbance on the reference ecosystem can be considered negligible. The Pleistocene Park core domain, which was sampled in the context of this study, has been influenced by grazing management since the establishment of the park in 1996, i.e. for about 25 years now. This information will be added to the methods text.

L91-93 and 101-103: Can you provide more precise information on e.g. vegetation composition and soil conditions prior to the onset of manipulating density of grazing herbivores? Something similar to Table 2 in Kwon et al 2016 would be a good start to support the central assumption that the sites were initially similar.

Unfortunately, there is no precise quantitative description of vegetation community structure for the Pleistocene Park area at the beginning of the landscape experiment in the 1990s. So we cannot provide information beyond the personal communication by our Russian project partners.

L103-104: For future reference, it would be good to mention to which plot numbers in Kwon et al 2016 the UGR-1 and UGR-2 sites in this study refer to. 2-0 and 2-2 are mentioned further in the discussion (L300) and should be mentioned here instead, but do not appear directly relatable to the denomination in Kwon et al 2016: does that mean Control-0 and Control-2?

Yes, the two sites selected for the presented study correspond to the sites labeled 'control-0' (UGR-1) and 'control-2' (UGR-2). This information will be added to the indicated text passage.

L108-112: Please clarify how many probes were used and where. From the phrasing I would have expected one probe per plot for temperature, and three probes per site for moisture, but from Fig.1 and the corresponding Results section it seems like there was only one probe in one of the GR plots and two in

UGR. Please also clarify what was considered 0cm depth in the water-logged tussock tundra (e.g. water-table, top of the tussocks, between tussocks), I assume it is the soil surface between tussocks but I shouldn't have to make assumptions in the Methods section. Please also mention the logging interval and the procedure used for producing the interpolated data presented in Fig.1.

It is correct that both UGR-1 and UGR-2 were equipped with soil temperature probes, while a single probe was used for all three GR-sites, due to their spatial proximity. The inter-tussock surface was considered as the reference height. All data was recorded at 1Hz during times of flux measurements. This information, including details on interpolation procedures, will be added to the modified version of the specified text passage.

L113-115: Please clarify what is meant by plausibility limits, which offsets were corrected and how. Please also provide further detail on the interpolation of soil temperature data: which variables other than air temperature and incoming radiation were used, and how. In addition, indicating the date and value of individual measurements in Fig. 1 would be helpful.

The soil moisture probes used in this experiment showed periods of enhanced signal fluctuation, which clearly could not be associated with a physical process. Accordingly, such periods were considered as low-quality data. Subsequently, noisy data in the observed soil moistures in 7.5cm, 15cm and 30cm depth was cleaned by selecting continuous intervals of low fluctuations as 'trustworthy'. Data points above and below visually determined plausibility limits around these intervals were removed from further analysis.

We considered this information to be of little relevance for the presented results, and accordingly decided to exclude it from the text to improve readability. The total number of individual observations, and days with flux measurements, corresponds to the number of flux measurements given in Table 1. We believe Figure 1 would be overloaded with information if such details would be added. Information on the flux interpolation procedures will be added (see also comment above).

L123-124: Please clarify whether a single relationship was used (if so, which one) or whether incoming radiation-PAR relationships were adjusted by incoming radiation classes.

We used a uniform conversion factor of 0.4730369 to convert between shortwave incoming radiation and PAR. This factor represents the average across the radiation bins presented in the cited reference, and was provided as a function within the R-package LakeMetabolizer (Winslow et al., 2016) that we used to perform this conversion. This respective information will be added to the manuscript text.

L139-140: I assume that not all measurements were 2 minutes long since that time is mentioned as a maximum. I would expect an arbitrary threshold in change in CO2 concentration was used to limit non-linearities from excessive CO2 buildup or uptake within the chamber, but could not find that value here or in Kwon et al 2016. Could you please clarify that part?

We did not define a threshold for maximum absolute change in greenhouse gas mixing ratios, temperatures or humidity within the chambers as a termination

criterion for a flux measurements. Based on previous field work experience with this chamber setup in the given environment, a target of two minutes was a practical compromise between collecting sufficient amounts of data, and avoiding biases due to non-linearities. Filtering out potential non-linear effects was included in the subsequent data processing, i.e. identification of periods with stable gradients in mixing ratios over time. The passage will be rephrased to clarify this issue.

L149-152: Is it correct to assume that these slopes were linear fits? If so, perhaps mention it at L150 instead of "steady", if not please clarify. In addition, a duration criteria must have been used for selecting the periods with a steady increase, please mention how long these periods had to be in order to be considered.

Yes, we assigned linear fits. We will change 'steady' to 'linear' as suggested by the reviewer to avoid confusion about this setting. The minimum duration of a linear period to be considered for gradient fitting was 40 seconds. This information will also be added to the text.

L152-155: I assume this is the reason for the uneven number of "utilizable" measurements in the different plots, if so please move the reference to Table 1 after this section instead of at L147. Please clarify in Table 1 what "utilizable" refers to. Please also mention somewhere why so few measurements were utilizable in UGR-2.

The reference to Table 1 will be moved as suggested, and the caption of Table 1 will be adjusted, including an explanation why UGR-2 is featuring less data than the other sites.

Appendix A: In line with my earlier comment about the number of probes used and their location, please clarify where the moisture / temperature data used for modelling comes from. Were GR-1, GR-2 and GR-3 models based on the same temperature data from the probe in GR-1? If so, was the data averaged between UGR-1 and UGR-2 or did these plots benefit from a distinct processing where they each had their own supporting data for the modelling?

As mentioned above, and already clarified in the text, both UGR-sites had their own soil temperature probe, since they were spaced approximately 50m apart, and were subject to different soil moisture regimes. For this reason, information measured directly next to the chamber frame was used for the modeling, and no averaging between the two sites took place. The GR-sites were clustered within a radius of approximately 6 meters, in seemingly homogeneous terrain. Therefore, additional soil monitoring did not seem necessary. This setup will also be clarified in the revised version of the Appendix.

L200: One of the assumptions of the Mann-Whitney test is that the observations are independent, but for both Ts and fluxes the data are time-series and are therefore not independent.

Motivated by the comment, we decided to adjust the statistical tests used for this purpose. In the revised manuscript version, instead of Mann-Whitney tests, a repeated measures ANOVA will be conducted prior to post-hoc pairwise t-tests and a correction of p-values with the Holm–Bonferroni method. In this way, we will provide evidence that the measurements are independent between days, and

correct the results of the pairwise comparisons for multiplicity problems. This approach will be used for fluxes, soil temperatures and radiation measurements.

L199-201 and throughout the text: The statistical tests results mostly present P values. I could not find precise guidelines of in-house rules regarding the presentation of statistical results in Biogeosciences, but I would suggest the test statistic to be presented as well, and the degrees of freedom. If this burdens the text too much, please consider a supplementary table.

Test statistics (Degrees of freedom, F-values, p-values for repeated measures Anova; corrected p-values for pairwise comparisons) will be provided in a new supplementary table, and will be referred to in the text.

L201: default t-test may not be appropriate with unequal variances, particularly so with unequal sample sizes as well. Please mention whether the assumptions for using a t-test were checked and met.

Statistics will be adjusted, and will be displayed as explained in the previous comments.

L207 and throughout the text: When presenting mean values, please provide associated uncertainties in the form of SD, SEM or CI.

Standard deviations will be provided when presenting mean values.

L223-224: Is there any data on this difference in air temperatures before the onset of the experiment? I.e. does this reflect initial differences or an effect of altered vegetation and soil conditions?

Unfortunately, there are no observations from the Pleistocene Park area before the onset of the grazing disturbance in the 1990s. Accordingly, there is no data-based proof that the differences in daily temperature amplitudes are an effect of grazing management. This, however, is not claimed in the manuscript.

L229-234: There is no mention in the Methods section of how these values were computed and it is unclear what the P values presented refer to, please clarify.

These results are related to the interpolation procedures for soil temperature data. We made tests with varying setups of input parameters, and also considered ranges for averaging intervals and time lags between air and soil temperatures, resp. As already explained above, we would like to exclude details on these procedures from the paper, since we consider this level of detail would compromised the readability. Therefore, we will also adjust this text passage, and remove most details referring to time lags.

L236: "clearly" is strong phrasing considering the absence of replication. Given the 100h lag at 15cm depth, effects of the change in weather pattern should only be visible in the last couple of days, at best, but one would expect deeper layers to have higher thermal inertia and therefore not to see an effect of the change in overall weather pattern by the end of the study period. In that respect, I do not think it is justified to carry out separate tests for the two weeks at this depth.

We agree with the reviewer that the data base is too slim to warrant such comparisons in the deeper soil layers. We will remove the word 'clearly', and also the last sentence that presents a statistical comparison of differences in the 2$^{nd}$ week.

L249-248: There is no mention in the Methods section of how or when thaw depths were measured or inferred from temperature data, please correct this.

Thanks for pointing out this oversight. A description of the measurement approach will be added to the methods section.

L252-277 and Figure 2: Judging by L199-200, the pairwise comparisons presented as letters in Figure 2 were computed by running 10 different Mann-Whitney tests per variable, plus one for the averaged values. If that is not the case, please describe this in the Statistics section, if that is the case, please clarify it in the Statistics section as well. In both cases, please indicate (how) were the P values adjusted for multiple comparisons. Beyond concerns about the Mann-Whitney test assumption of independence of observations expressed above, I would advise running an omnibus test prior to post-hoc pairwise comparisons. With a balanced design, a repeated-measures ANOVA could be a correct way to account for dependent observations within a plot. Considering the central role of flux data in this manuscript, their statistical treatment should be improved.

The previous tests based on Mann-Whitney have been replaced by an alternative approach. Please see the above comment for details.

L256-258: It is unclear what "flux rates" refers to in the first part of this sentence: NEE, Reco?

The statement was completely referring to GPP rates. The passage will be re-formulated to clarify this issue.

L286-287: When were the collars installed at the GR site? Presumably after setting up the wooden fences to prevent trampling by the herbivores, but please mention this here or in the Methods section.

The collars at the GR sites were indeed installed only days before starting the field experiment in 2019. This will be added to the discussion section.

L293-295: Why not mention tussock-forming plants here? As far as small-scale heterogeneity is concerned it seems odd not to mention one of the main ecosystem engineers of these systems.

As suggested by the reviewer, tussock-forming plants will be added to the list of factors contributing to fine-scale structuring of ecosystems.

L308-309: It might be good to remind here that when comparing CVs of GR and UGR one should keep in mind GR having 50% more plots and ~100% more measurement points.

A statement referencing this imbalance in the GR vs. UGR datasets will be added to the end of the paragraph.

L324: See above at L108-112, this is not a reminder and this information should be stated more explicitly in the Methods section.

Since the methods section will be extended according to the remarks by the reviewer listed above, the text passage referred to here is now actually a 'reminder' to the methods, and therefore will be kept as is.

L329: I assume "the actual measured values" refer to Reco, but please clarify.

This statement will be removed from the manuscript text.

L351-355: It would be good to mention examples of which such operations might be confounded with the effects attributed to the increased grazing herbivore density.

Examples of management operations will be added to this sentence.

L357-359: See my earlier comment about L91-93 101-103, in absence of more detailed data such photographs may be an interesting supplementary display item.

We have shown such photos already in our response to reviewers' comments when updating the manuscript that was ultimately accepted as a Discussions paper. While these photographs give a qualitative impression how the Pleistocene Park grasslands have changed over the past 20 years or so, it is still hard to draw conclusions since even pictures taken from exactly the same perspective (such as the one shown for the park) suffer from differences in season, and incoming light. We would therefore prefer NOT to show such pictures in the appendix of our manuscript; however, since the picture material is obviously available, we would leave this decision to the editor.

L394: "mostly likely" should be "most likely", but the phrasing is a bit strong for an hypothetical future development, which to date is in contradiction with the observations as shown in Fig 1. While I understand the hypothesis of a cooling of the soil and grazing-induced protection of permafrost in Pleistocene Park, it is hard to ignore that Fig 1 shows an almost twice as deep active layer thickness in the grazed site. Either the hypothesis is correct but the sites differed drastically in active layer thickness prior to the experiment, or the effects observed after 22 years of manipulation contradict the expected consequences of the hypothesis. A transient regime is possible but less parsimonious, and "most probably" or "most likely" are too strong for that to my taste.

We agree that future trajectories can only be speculated upon, and that the statement therefore should be toned down. We will change 'most likely' to 'potentially'.

L404-406: Liquid water has a fairly high thermal conductivity, a comparison between values for a compacted soil and a water-logged soil could be useful information here.

A statement on the comparison of thermal conductivity between soil minerals and water will be added to this phrase.

L408-410: It is hard to say for 25cm depth since the data is not shown, but for 35cm it would be good to remind that the observed difference in soil temperature is lower or similar to the observed difference in air temperature.

We are not sure how to interpret this statement. It is obvious that dynamics in soil temperatures are muted in comparison to those in the atmosphere. This also applies to differences observed between sites. We currently do not see how we could include this into the discussion at the referenced section. We would certainly be interested in incorporating this comment into the revisions, but the reviewer would need to provide more details on what s/he has in mind.

L414-415: In line with the previous comment, this sentence could be complemented by starting it with "Barring differences prior to the onset of the experiment".

The suggested statement will be added to the beginning of the last sentence of this paragraph.

L416-419: Considering that no difference was observed in growing season NEE and that in presence of grazers, a larger fraction of NPP is removed by herbivory, this argument should be substantiated with above- and below-ground plant biomass measurements or a complete C budget. In their absence, it is speculative and because this is not central to the reasoning, I would suggest removing it.

We agree that our current result cannot support the given statement, and therefore will remove it from the revised manuscript.

L419-420: See above at L394, this is speculative and in direct contradiction with data presented in Fig. 1.

The statement will be removed.

L453 and 456-457: This is speculative, please use less strong phrasing.

All statements will be toned down as the reviewer suggested.

Technical corrections

L27: change sentence order

We will change the sentence structure.

L57: facilitates -> allows

The word will be exchanged accordingly.

L116-119: I would suggest using GR and UGR rather than Pleistocene Park and Ambolikha for consistency.

We will add GR and UGR, resp., to the site description for consistency.

L165-166 and L200: Rstudio is only a GUI software to R and does not do calculations. Please move the mention to the software used to the end of the statistics section, and provide adequate reference including R, the version number and the appropriate citation (e.g. R Core Team. R: A language and Environment for Statistical Computing. (2021)

Citations will be added for R, replacing R-Studio.

L200: "(?)"?

Due to a formatting error, the chosen reference was not displayed properly. This will be fixed.

L226 – Figure 1: I would recommend making two separate panels out of panel (a). I do not think the y-axis break simplifies the figure, and the factor 5 change in axis scale would be more obvious that way.

L437-439: Please consider rephrasing, the current syntax poses "increases in primary productivity" as an explanation for increased GPP.

The sentence will be rephrased.

L480: "differences in NEE were not pronounced" -> "no differences in NEE were found"

The statement will be rephrased according to the reviewer's suggestions.

Appendix A, L494-496: This sentence would be easier to understand if the information was split across several sentences, please rephrase it.

The sentence will be split up, and slightly rephrased, to make it easier to understand.

**References cited:**

Kwon, M. J., Heimann, M., Kolle, O., Luus, K. A., Schuur, E. A. G., Zimov, N., Zimov, S. A., and Göckede, M.: Long-term drainage reduces $CO_2$ uptake and increases $CO_2$ emission on a Siberian floodplain due to shifts in vegetation community and soil thermal characteristics, Biogeosciences, 13, 4219-4235, 2016.

Winslow, L., Zwart, J. A., Batt, R. D., Dugan, H., Woolway, R. I., Corman, J., Hanson, P. C., and Read, J. S.: LakeMetabolizer: An R package for estimating lake metabolism from free-water oxygen using diverse statistical models, Inland Waters, 6, 622-636, 2016.

**Author response to interactive comment RC2 submitted on Sep 01, 2021**

In the document below, the reviewer comments have been copied from the original review and are shown in black font, while the author comments have been added in blue.

General comments

The authors provide a nice introduction into herbivory impacts on permafrost ecosystems. The study provides a very interesting insight into ecosystem changes under grazing pressure. The data set used is a measurement series of NEE and Reco, measured for two weeks at a grazed and an ungrazed site with several replicates. The observed flux changes in CO2 and CH4 are well described and put into relation with animal activity, such as soil compaction and drying, which shows a significant reduction in CH4 emissions from grazed sites.

The methods used are suitable to use for the provided explanation of these effects, however, the method description itself should provide more detail on the approach. There are several further topics arising from this study, such as the influence of vegetation species on fluxes and how fluxes change throughout different seasons. It would be great to have more comparison to other studies regarding this.

The methods description will be extended at several places, with details being described in our answers to the specific comments below, and also in our reply to the comments of Reviewer 1. We decided against discussing seasonality effects, since our dataset clearly does not cover seasonal variations, therefore we cannot contribute new information to this topic. We decided to also change the title of the manuscript to reflect that this is a short-term study within the peak growing season.

There is a minor lack of context regarding the general hypotheses of the Pleistocene Park experiment as to why the findings from this study suggest a different effect of animal grazing than previously hypothesized by Zimov et al. (2005). The findings should also be discussed in relation to those hypotheses.

We assume the reviewer refers to the fact that we observed warmer soils, and deeper thaw depths, at the grazed site, compared to the ungrazed reference. We discuss the potential effects of changes in heat capacity and thermal conductivity in depth in Section 4.5. The Zimov hypothesis postulates a lowering of the annual soil temperatures as a response to snow compaction in the winter, which does not contradict our findings of temporally warmer soil conditions during the peak growing season, but rather adds more detail to the original general claim without specifying seasonal dynamics. Since the overall assessment mostly depends on potential grazing influences on soil temperature shifts in the non-growing season, which we did not capture with our measurements, we decided to leave this out of the discussion, since we could only speculate. The remaining hypotheses on the Pleistocene Park concept are actually confirmed by our observations,

although we certainly do not consider our results as 'proof', since their temporal and spatial representativeness still needs to be evaluated. Still, based on this reviewer's comments, we picked up the hypotheses again in our conclusions.

Specific comments

Please consider making the data accessible via a scientific data repository.

Upon publication of the presented manuscript we plan to upload the data to a publicly accessible repository.

L89: Please add a map indicating the sampling sites.

We will add map and photo material showing the sampling sites in an additional appendix part of the revised manuscript

L91: There is a new paper by Reinecke et al. (https://doi.org/10.1038/s41598-021-92079-1) dealing with the Pleistocene Park vegetation in more detail, which you should consider here.

The paper by Reinecke et al. will be referenced; however, it does not provide a specific vegetation community structure for the Chersky study area, instead it summarizes across study sites in Yakutia how grazing by different species induced shifts in plant communities.

L151: Please describe the bootstrapping approach in more detail (number of iterations etc.).

Additional details on the bootstrapping approach will be added to the text.

L206: How did you test for significance?

We decided to adjust the statistical tests used for this purpose. In the revised manuscript version, instead of Mann-Whitney tests, a repeated measures ANOVA will be conducted prior to post-hoc pairwise t-tests and a correction of p-values with the Holm–Bonferroni method. In this way, we will provide evidence that the measurements are independent between days, and correct the results of the pairwise comparisons for multiplicity problems. This approach will be used for fluxes, soil temperatures and radiation measurements. This approach will be explained in the Methods section and not in the results section.

Figure 2: For $CH_4$, it should be clearly stated that these are emissions only. Using "fluxes" suggests a bi- or omnidirectional gas exchange.

In principle, these chamber techniques do measure omni-directional gas exchange. When closed, the hoods capture all gases that leave the soil, at the same time they also detect when gases are removed (e.g. $CO_2$ reduction through photosynthesis, or $CH_4$ reduction through oxidation). At very dry sub-plots of the study sites, negative $CH_4$ flux rates have been detected. Still, we can change 'CH4 fluxes' to 'CH4 emissions', since in this case all study plots were actual sources for the gas.

Table 2: I assume "ns" means "not significant"? Please make the caption overall more clear. Also, please add something like "ungrazed sites (UGR-1 and -2) and grazed (GR-1, -2 and -3)" to the title of this table. I suggest, for uniformity, to switch axes of this table to make it similar to table 3.

The caption and also the subscript will be amended to clarify the data being presented. The axes of the table will be switched to align with the format shown in Table 3, as suggested by the reviewer.

L295: What about previous disturbances of the soil itself, especially in the active layer with freeze-thaw cycles? Please consider this in your manuscript

Freeze-thaw dynamics will be added as an aspect leading to small-scale spatial heterogeneity in tundra landscapes.

L357: These pre-existing site differences are very likely, taking the distance between the sites into account. Especially the differences in thaw depth (greater thaw depth at UGR) are opposing the general hypothesis of large animal impact on permafrost ground as a conservation mechanism, which is said to mainly originate from snow compaction in winter. Maybe you should elaborate or highlight these a little more and discuss why your findings might differ from named hypothesis.

Both sites used to be seasonally water-logged tussock tundra ecosystems within the Kolyma River floodplain, and in this context a distance of 15km should not be relevant. Grazing disturbance within Pleistocene Park has altered vegetation community and hydrologic status, but as also outlined in our response to the comments by Reviewer 1 we unfortunately cannot provide data-based proof on the similarity of site conditions in the pre-treatment era. Within such ecosystems, even minor differences in elevation, slope or soil conditions can impact the carbon and energy cycles, but such differences can also occur within scales of just a few meters. The only approach to circumvent this small-scale spatial variability would have been to select many more sites hoping that their variance would asymptotically approach the true variance independent of horizontal distance. Given the logistical challenges and site access, this approach was not pursued.

It is correct that an increased thaw depth, and also overall increased soil temperatures, are not in agreement with the overall hypotheses regarding herbivore grazing effects on Arctic tundra ecosystems. However, since our dataset only covers a short snapshot in time during the peak growing season, these observations cannot be interpreted with regards to year-round conditions. As previously mentioned, our findings do not contradict or disprove the original expectations, but rather add a desired level of detail for the peak growing season. It can be speculated that wintertime and early growing season temperatures were actually lower in the grazed section, compared to the ungrazed references, and that a combination of reduced heat capacity (due to lower soil moistures) and increased thermal conductivity (due to a reduced organic top soil layer, and compacted soils) led to a rapid warming in the managed ecosystems. Both effects are discussed in detail in Section 4.5 in the manuscript, which will be modified in the revised version to clarify our arguments (see also responses to Reviewer 1).

Figure A1: Please provide letters for each graph (e.g. as in figure A3). Also, adding the equation for each regression curve to the corresponding graph would be good.

We will change the figure accordingly.

Figure A2: Please see the comments on figure A1.

We will change the figure accordingly.

Figure A3: Please add the equations for each regression curve.

We will change the figure accordingly.

Figure A4: Please provide headlines for a), d) and g). Also, it should say somewhere in the graph (not only in the caption) that the graphs show CH4 emissions.

We will change the figure accordingly.

Technical comments

Please make "C-Fluxes / C-fluxes / C fluxes" consistent throughout the paper. Maybe consider replacing flux considering my earlier comment

We will change the expression to 'C-fluxes' throughout the text.

L99: Please put Betula nana in italics and capitalize, since it's a species name. Also, please change "willow spec." to "Salix sp."

The entries will be changed accordingly.

L100: Please change "lugens" to "C. lugens".

Will be done.

L166: R Studio is just the main software. Please provide the used packages.

This will be changed accordingly (see also comments by Reviewer 1).

L170: Suggestion: "…not uniform across plots even at one site…"

This will be changed as suggested.

L200: There is a leftover "?" in this line. Also, the test should be named "Mann-Whitney-Utest".

The statistics have been adjusted, so changes to this phrasing are not needed anymore.

Table 2 caption: inconsistency in * and spaces, please adjust

This will be adjusted.

Line 283: please capitalize

We actually do not see a word in need of capitalization in this sentence. If something still needs to be changed, we would need more precise comments.

**Author response to interactive comment CC1 submitted by Cole Brachman on Jul 20, 2021**

In the document below, the reviewer comments have been copied from the original review and are shown in black font, while the author comments have been added in blue.

The manuscript aims to determine the role of grazing in carbon cycling through $CO_2$ and $CH_4$ gaseous fluxes in wet tundra habitat by the means of the large-scale herbivore reintroduction experiment of Pleistocene Park. The authors measured ecosystem respiration (Reco), Net Ecosystem Exchange (NEE) and $CH_4$ using chamber methods and a flow through gas analyzer over seventeen days in five different plots distributed over two sites, one for the grazed (GR) condition within Pleistocene Park and one for the ungrazed (UGR) condition located nearby to the park. Gross Primary Productivity (GPP) was also calculated from Reco and NEE. The fluxes were interpolated based on the chamber measurements, air and soil temperatures, and soil moisture conditions over the measurement period. There were differences in the fluxes between the site conditions, which were primarily attributed to grazing having a drying effect on the GR sites. These initial findings, if further verified with additional measurements as outlined below, could result in some important implications for the role of grazers on the tundra landscape. Overall, this paper hints at some very interesting connections between carbon cycling, environmental conditions, and grazers but require some additional measurements to support the bold claims as they are currently in the manuscript.

Major comments:

The data are not enough to support the claims being made in the manuscript. The limited number of independent measurements and an unequal sampling design undermine the conclusions reached about the relationships. 17 days of measurements give an accurate estimate of the fluxes over that period, but do not necessarily represent the whole growing season. It is mentioned in the paper that these should be treated as a snapshot in time (especially for the GR plots), however, I do not believe the main takeaway points as they are written are properly taking that caveat into account which can result in some miscommunication of the strength of the findings. Additionally, only having two plots in the UGR condition, and only measuring those plots four times (4 days compared to 9 days for the three GR plots) makes accurate comparisons between the treatment types difficult for the full measurement period.

We are aware that, based on the limited available database, particularly quantitative results are associated with considerable uncertainties, but we are confident that this fact is well reflected in the discussion of the material. To further emphasize the limited database and temporal coverage, the title will be modified in the revised manuscript version, now reading "Grazing enhances carbon

cycling, but reduces methane emission during peak growing season in the Siberian Pleistocene Park tundra site". Also, a new statement will be added to the end of the abstract (see comments to reviewer 1).

At the same time, we are certain that our results capture the dominating qualitative shift in ecosystem characteristics and carbon cycle dynamics that follow a decade-long, intensive grazing disturbance in these very sensitive Arctic wetlands. Even though our carbon flux estimates cannot be proven to be representative for larger areas outside of the flux footprint, we believe that our study provides valuable and novel insights into the impact of such management practices, and their application as a potential tool to protect Arctic permafrost from degradation under climate change.

Many studies covering novel, uncharted scientific territory in regard to method and/or location may be associated with a larger uncertainty compared to repeating established methods at previously studied locations. While we do not intend to discount the scientific contribution and merit of the latter, it may be rather incremental. From all possible forms of scientific inquiry, our abductive method is more speculative, but we strove to provide and include all information at our disposal in support of our results and claims. We will further strengthen this aspect by adding a more detailed discussion of the shortcomings, as also documented in our responses to the reviewer comments.

The two selected UGR plots had large differences in their GPP and NEE measurements and may not be a good representation of these sites. Selecting additional plots from the 10 previously established UGR plots for measurements would help to more accurately determine average flux values. The individual UGR plots are also showing very similar fluxes as the GR plots, but not consistently (see table 3). For instance, UGR 1 have similar GPP and CH4 as the GR plots, while UGR 2 seems to bring down the average GPP in the UGR plots. In addition, the UGR plots were not measured on the same days. This clearly demonstrates how the low replications undermine their conclusions.

We agree with the reviewer that a database comprising only two sampling sites cannot provide a statistically sound representation of an observation site featuring fine-scale variability. However, it was clear from the onset of the experiment that a data coverage of just 2.5 weeks could not provide a comprehensive assessment of grazing impacts. This was never our intention, as mentioned already above, and this is also clearly stated in the manuscript text.

The rationale behind our site selection was already discussed and explained at length in our responses to the comments of Reviewer 1, these statements are therefore repeated here:

The decision to work with only two reference sites (UGR) plots at the Ambolikha site was based on practical considerations. In principle, we could have used up to 10 sampling locations which had been established in earlier experiments. However, plots were spaced 25m apart, meaning that the observation system has to be moved between sites when switching locations, as opposed to the GR sites, which were co-located within a narrow radius. Spending time for moving the system implies less time for actual measurements, which is why we wanted to reduce it. Therefore, the choice was made to only sample two sites.

We realized that the description in the submitted version of the manuscript chosen to justify the UGR site selection was somewhat misleading. While fluxes at the two selected were actually indeed close to the mean fluxes across the transect, our choice was rather motivated by the ecosystem structure. While we cannot give more precise information on the GR sites before grazing started, we know from personal communication that the managed area used to be a waterlogged tussock tundra. Out of the 10 plots that were available at the UGR site, six are dominated by cotton grasses (Eriophorum), with few or no tussocks present (see Figure 8 from Kwon et al., 2016, copied below). Two more sites (IDs 4 and 5) were placed on a small ridge, and were therefore significantly drier, and dominated by shrubs. We therefore selected the only two locations, IDs 0 and 2 in the control section, featuring the desired vegetation structure for investigating the effects of grazing. Studies with a different scope may have enabled a random site selection to improve estimates of uncertainty due to site-specific bias.

[Figure]

The authors actually cannot follow the rationale that fluxes across chambers need to be measured on the same measurement days in order to be comparable. This is clearly not practical when sampling sites are located far apart, and even impossible for experiments that include large numbers of sampling spots that are regularly revisited. In our setup in the Chersky region, taking instrumentation from the grazed to the ungrazed study site, or vice versa, would have taken about two hours – precious time that we preferred to rather invest into actual measurements. As long as there are no systematic and fundamental differences in weather conditions between measurement days, we concluded it is fully sufficient to aim at capturing fluxes across a wide range of environmental conditions at each site in order to allow fitting response functions.

Site differences between the GR and UGR plots make it difficult to determine if the differences in fluxes are actually due to grazing effects and not moisture itself. Stronger evidence of the GR plots being water-logged throughout the growing season ~30 years previous, and that the drying of the site is due to grazing, is necessary to solidify the link between grazers and fluxes. Alternatively, flux measurements on wetter areas in Pleistocene park, and dryer areas in the UGR site may help disentangle the effect of moisture from the effects of grazing.

We would have liked to include some data-based evidence in the manuscript that demonstrates that both sites had similar pre-treatment characteristics, and only started to diverge with increasing grazing pressure at GR over the past decades. However, direct measurements of ecosystem characteristics within Pleistocene Park from the 1990s or before are not available, including soil moisture assessments that could help to compare soil hydrology over the past decade in connection with the grazing management.

Also remote sensing products such as e.g. LandSat time series, which are available in several scenes per year since 2000 for both study areas, turned out to be ill-suited for this particular purpose. As we obviously lack in-situ observations from the pre-treatment stage, we resorted to discussing this aspect thoroughly in the manuscript while mentioning that potential differences in pre-treatment conditions may add a systematic bias to the differences in carbon fluxes obtained from our chamber measurements. As already mentioned in our response to Reviewer 1, in the revised version of the manuscript we will further tone down some statements in the abstract, and add an additional statement to the end of the abstract:

"Our results indicate that grazing of large herbivores may promote topsoil warming and drying, this way effectively accelerating $CO_2$ turnover while decreasing methane emissions. Lacking quantitative information on the pre-treatment status of the grazed ecosystem, however, these findings need to be considered as qualitative trends, while absolute differences between treatments are subject to elevated uncertainty. Moveover, our experiment did not include autumn and winter fluxes, and thus no inferences can be made for the annual NEE and $CH_4$ budgets at tundra ecosystems."

Minor comments:

L 21: "Based on expert assessment", please delete.

The quote 'expert assessment' was actually taken over from the Schuur et al. (2015) reference quoted in this sentence. However, we agree that this statement may be misleading, and therefore re-formulated to "Based on several independent approaches, it is estimated that 130 to 160 Gt C could be released by 2100 .."

L 53: The drawbacks of measuring fluxes only in the growing season were mentioned, however, this study also only measured fluxes during a subset of the growing season. Consider leaving this to the discussion section as the reader expects some mention of a whole-year upscaling when it is mentioned early on in the introduction.

This reference was also criticized by the other reviewers, and is addressed in more detail there. In short, we agree that it may be misleading to refer to year-round fluxes in the introduction when our study does not deal with them, and therefore removed this sentence.

In the introduction, there are multiple mentions of shrubs and the effect of shrubs on C dynamics (possibly due to a large amount of the reference studies coming from Scandinavia and focusing on reindeer browsing), but your sites are dominated by graminoids. I would suggest reframing the introduction to focus more on the effect of graminoids on C dynamics and their interaction with large herbivores. This is also not much elaborated in the discussion, and the introduction as it reads now give the wrong expectations on the manuscript.

We agree with the reviewer that shrubs are not a dominating factor for our experiment carried out in the Kolyma lowland region, though shrubs certainly are an important element for the vegetation composition within the floodplain. However, the term 'shrub' is mentioned exactly three times in the introduction: once in a general section on Arctic climate change that is not focusing on grazing, a second time when citing potential influences of herbivore grazing on tundra vegetation, and a third time when listing hypotheses postulated for the Pleistocene Park experiment. The most important of these statements, i.e. the second one, is directly followed by the sentence "Grazing has been shown to promote certain Carex species that produce a high belowground biomass, ..". We believe our use of the term 'shrub does not raise incorrect assumptions or expectations' in the reader and thus is not misleading.

In the discussion, Section 4.4 which focuses on "Grazing Impacts on Vegetation" actually strongly focuses on graminoid species, and their relationship to grazing. We therefore disagree also with the claim that graminoid interaction with herbivores is not much elaborated on. To cite some examples:

- almost all sedge-tussocks were in a state of decay, or had disappeared almost completely. In place of them or between their remnants, many single plant tillers (mainly Carex spec. and Calamagrostis langsdorfii) grew.
- the transformation from tussocks to grass mats by grazing, accompanied by a strong increase in belowground biomass, was already observed for montane biomes
- Some sedges found in Arctic environments, such as Carex aquatilis, were shown to benefit from muskox-grazing, since they feature strong root production and the ability to produce dense grass tillers, and therefore more easily recover from grazing
- Accelerated urea-nutrient uptake by living plants has been reported for upland tundra (Barthelemy et al., 2018), where graminoids were more efficient in using these resources compared to shrubs.

Suggest renaming the plots from grazed (GR) and ungrazed (UGR) to heavily grazed (HGR) and ambient grazed (AGR), respectively, unless there are no populations of grazing herbivores on the landscape at the ambient site (no information provided).

We already changed the site description accordingly, based on a comment by Reviewer 1: "Our undisturbed reference ecosystem is a tussock tundra site situated

about 15km south of Chersky on the floodplain of the Kolyma River. Due to the very low natural abundance of grazing herbivores in the region, the influence of grazing disturbance on this dataset can be considered negligible.". We will therefore stick to the site descriptions GR and UGR.

L 200: Mann-Whitney U tests were brought up in the statistics section but I could not find the results or a figure on these tests. Since these measurements also are repeated measurements, you need to provide evidence that they are independent between days (your statistical unit) or perform statistical test considering the repeated measures.

We decided to adjust the statistical tests used for this purpose. In the revised manuscript version, instead of Mann-Whitney tests, a repeated measures ANOVA will be conducted prior to post-hoc pairwise t-tests and a correction of p-values with the Holm–Bonferroni method. In this way, we will provide evidence that the measurements are independent between days, and correct the results of the pairwise comparisons for multiplicity problems. This approach will be used for fluxes, soil temperatures and radiation measurements.

L 306-311: Coefficients of Variance (CV) were discussed to determine if the heterogeneity between plots were in an acceptable range. However, when compared to the paper cited as a reference for this metric (Davidson et al. 2002), the present study has half the number of total plots they are assessing over which could be a factor in the low values found. The Davidson et al. (2002) paper also suggests a formula for determining the number of measurements needed to ensure a decent variance around the mean, which could be a useful way to determine if the number of measurements taken are representative or if more measurements are needed. In addition, it is unclear what measurements the CV is calculated on. It should be the daily data, 4 measurements for UGR and 9 for GR.

We will add an additional sentence to this paragraph to highlight the fact that the database was not equally distributed between GR and UGR sites: "However, one has to keep in mind that the GR sites feature more plots than UGR, and also a higher number of observations, both of which may influence a comparison of derived CVs."

Equation 3, which corresponds to interpolating Reco from UGR plots according to section 4.2 (lines 320-322), includes the data from GR-3. The interpretation of data from the GR plots therefore differ from each other, and GR-3 is interpolated more accurately with the same formula as that for the UGR plots. This was mentioned on line 326 stating that the measurements are not representative across the GR plots, which poses problems for the final conclusions drawn regarding these plots.

Equation 3 was indeed used to interpolate Reco for both the 2 UGR sites and the GR-3 site. This is stated in lines 326f in the Discussion paper: "For that reason, at GR-3 also Tair was used to interpolate Reco, since …". Our interpretation of the fact that we find different response functions for Reco across the GR sites is that there is obviously some micro-scale variability within a seemingly homogeneous ecosystem, and that we were able to capture this variability through our three sampling sites. GR-3 appears to be slightly drier than the other 2 GR sites, which is also reflected in the CH4 flux rates. However, we do not see how this poses a problem for the conclusions drawn in our study.

L 364-373: Is it possible to tie these vegetation changes into the differences in measured fluxes more directly? Maybe a reference on fluxes from tussocks vs. grass mats?

Unfortunately, we are not aware of reference studies that directly compare the flux rates between tussocks and grass mats under the same environmental conditions. Some of our own work on the Ambolikha site compared fluxes from tussock-dominated patches to those with dense cotton grass meadows, which is e.g. reflected in the figure from Kwon et al. (2016) copied above. Here, the cotton grass meadows (Eriophorum plots) featured higher GPP and lower Reco, compared to the Carex-dominated plots. However, this is not precisely a good reference for 'grass mats' that may develop under grazing pressure.

L 373-375: Were the addition of CO2 and CH4 from grazers themselves factored into any calculation of total fluxes from the sites?

No, direct emissions from herbivores were not considered in our estimates.

Clarification of the prevalence of these wet tussock tundra sites within and outside of Pleistocene Park would be a useful addition when visualizing how these results may affect the larger arctic region.

Current pan-Arctic vegetation maps are not yet detailed enough to differentiate wetland features such as e.g. wet tussock tundra. For example, in a recently published study by Olefeldt et al. (2021), Arctic wetlands were merely separated into 'permafrost wetlands' and 'permafrost bogs', and even this can be considered a big advance from aggregating all kinds of wetlands into a single vegetation class.

L 402: This sentence needs a reference at the end.

We will add the study by Göckede et al. (2017) as a reference here.

L 403: "only very inefficiently", consider revising.

The sentence we be re-written.

L 731 reference for Zimov et al. 2012, seems to have the incorrect initials for one author (F. S. Chapin).

This typo will be corrected in the revised manuscript.

**References cited:**

Göckede, M., Kittler, F., Kwon, M. J., Burjack, I., Heimann, M., Kolle, O., Zimov, N., and Zimov, S.: Shifted energy fluxes, increased Bowen ratios, and reduced thaw depths linked with drainage-induced changes in permafrost ecosystem structure, Cryosphere, 11, 2975-2996, 2017.

Kwon, M. J., Heimann, M., Kolle, O., Luus, K. A., Schuur, E. A. G., Zimov, N., Zimov, S. A., and Göckede, M.: Long-term drainage reduces CO2 uptake and increases CO2 emission on a Siberian floodplain due to shifts in vegetation community and soil thermal characteristics, Biogeosciences, 13, 4219-4235, 2016.

Olefeldt, D., Hovemyr, M., Kuhn, M. A., Bastviken, D., Bohn, T. J., Connolly, J., Crill, P., Euskirchen, E. S., Finkelstein, S. A., Genet, H., Grosse, G., Harris, L. I., Heffernan, L., Helbig, M., Hugelius, G., Hutchins, R., Juutinen, S., Lara, M. J., Malhotra, A.,

Manies, K., McGuire, A. D., Natali, S. M., O'Donnell, J. A., Parmentier, F. J. W., Räsänen, A., Schädel, C., Sonnentag, O., Strack, M., Tank, S. E., Treat, C., Varner, R. K., Virtanen, T., Warren, R. K., and Watts, J. D.: The Boreal–Arctic Wetland and Lake Dataset (BAWLD), Earth Syst. Sci. Data, 13, 5127-5149, 2021.

---

## Referee Report (RR1)

**2nd review of bg-2021-110**

**Grazing enhances carbon cycling, but reduces methane emission in the Siberian Pleistocene Park tundra site**

**General comments**

In this revised version of the manuscript, the authors excellently managed to clarify their study design for comparing carbon emissions and uptake between grazed and ungrazed Arctic tundra sites.

Detail additions in both introduction and methods will help readers to understand the study's intention and limitations. These limitations are picked up again in the discussion, and discussed in sufficient detail.

The additional work put into graphical design improves readability of the graphs and understanding of the "read thread" drastically.

Adding a paragraph on the original hypothesis to the conclusions makes this paper a well-told story with interesting but also very specific findings.

**Specific comments**

Please consider making the data accessible via a scientific data repository.

**Technical comments**

Line 90: There's a missing space between 15 and km.

Table 2: There is still some inequality in spacing of the asterisks in the table description.

Line 741: There's a typo in the reference to Myers-Smith et al., where the I in "Macias-Fauria, M." should not be capitalized.

**Review criteria:**

*Does the paper address relevant scientific questions within the scope of BG?*
Yes

*Does the paper present novel concepts, ideas, tools, or data?*
Yes

*Are substantial conclusions reached?*
Yes

*Are the scientific methods and assumptions valid and clearly outlined?*
Yes

*Are the results sufficient to support the interpretations and conclusions?*
Yes

*Is the description of experiments and calculations sufficiently complete and precise to allow their reproduction by fellow scientists (traceability of results)?*
Yes

*Do the authors give proper credit to related work and clearly indicate their own new/original contribution?*
Yes

*Does the title clearly reflect the contents of the paper?*
Yes

*Does the abstract provide a concise and complete summary?*
Yes

*Is the overall presentation well structured and clear?*
Yes

*Is the language fluent and precise?*
Yes

*Are mathematical formulae, symbols, abbreviations, and units correctly defined and used?*
Yes

*Should any parts of the paper (text, formulae, figures, tables) be clarified, reduced, combined, or eliminated?*
No, all good

*Are the number and quality of references appropriate?*
Yes

*Is the amount and quality of supplementary material appropriate?*
Yes

---

## Author Response (AR2)

**Author response to comment submitted by both reviewers to the revised version of this manuscript**

In the document below, the reviewers' comments have been copied from the original reviews and are shown in black font, while the author comments have been added in blue.

**Comments by Reviewer #1**

General comments

I believe the changes made by the authors since the last version have substantially improved the manuscript, most importantly on the statistical analysis and the description of the study sites and rationale for their choices. I only have minor comments/suggestions that would further improve this manuscript prior to publication, which I detail below.

Thanks a lot for this very positive evaluation!

1. A related preprint on the Pleistocene Park is under discussion in the same journal (Windirsch et al., https://doi.org/10.5194/bg-2021-227) and the corresponding author is also involved in that study. For further comparisons, it would be good to mention how the GR site relates to the sites in that study (it seems to be fairly close to DB-IN?), and to discuss the apparent contradiction between the observed effects on thaw depth in the two studies. The distinct "control" sites are certainly responsible for the different findings, but since Windirsch et al. observe a thicker active layer in both lowland and upland sites in absence of large herbivores, some discussion of this would be of interest to the readers.

The work presented by Torben Windirsch and colleagues is indeed closely related to the work presented in our manuscript. Parts of the field work for both studies was carried out together, and both sides supported each other in carrying out measurements, and/or taking samples.

As correctly noted by the reviewer, the site DB-IN (in the revised version of Torben's manuscript re-labeled to B3) is indeed located very close to the GR sites where fluxes were measured for our study. The horizontal distance between the position of our chamber frames and their soil sampling spot was approximately 12 – 15m. Windirsch et al. measured a thaw depth of 38 cm at a single location and time for this site, while the values in this manuscript vary between 39 – 58 cm over three different sampling spots and a period of 10 days.

The seemingly different values for this sampling site itself can be explained by sampling time: Torben Windirsch analyzed this site a couple of days before our first measurements, while we were still setting up. Given the steep trends in thaw depths, e.g. reflected in Fig. 1b in this manuscript, and also the spatial variability as reflected by the 3 sampling spots used herein, the values given in both studies therefore correspond well with each other.

The differences in thaw depth between this site and the respective references in both papers can be explained by wetness levels. In Torben's study, the B3 site is labeled as a 'wet area of the thermokarst basin', while the 2 grazed references sites do not carry the 'wet' label. Accordingly, while the reference sites in the Windirsch study are actually drier than B3, in our study the references (UGR) are permanently water-logged, and therefore much wetter than the GR sites. The effects of soil moisture changes on heat capacity and heat conductivity are discussed in-depth in our Section 4.5 already. In the paper by Windirsch et al., the potential impact of wetness is already mentioned in their methods section 3.1: 'Also, flooding regime (seasonal or occasionally) is different between our sites and might have some effects on the soils.' Within their discussion Section 5.2, they then mention that soil moisture may have a profound impact on soil organic storage, and a co-existing influence of grazing and soil moisture should be investigated more closely in follow-up studies.

To reflect the correspondence between the two related studies, we changed the last part of the second paragraph of Section 4.5 on 'Grazing impacts on soil properties' in our manuscript. We removed the sentence 'Barring differences prior to the onset of the experiment, these studies suggest that the differences in soil properties between GR and UGR may be predominantly attributed to grazing pressure.', and replaced it with this new passage:

"The important role of soil moisture conditions is also highlighted in the results by Windirsch et al. (2021), who investigated places with different grazing pressure within the same thermokarst basin in Pleistocene Park where our GR sites were located. The drier locations showed a deeper thaw depth in their study, even though the grazing pressure at these sites was actually lower. In accordance with Windirsch et al. (2021), we therefore conclude that both grazing pressure and soil moisture differences hold the potential to substantially influence the soil properties, and their co-existing influence needs to be tested in further experiments."

2. The thermal conductivity of minerals is a possible explanation for the altered thermal regime, however the reference chosen to support this focuses on soils with less than 3% organic matter and states that results could be drastically different for more organic soils such as peat. Considering such data is not presented in this study, one can use the DB-IN data in Windirsch et al. Fig 2 as a close data source, which shows 10-25% C in the top 50 cm, which would amount to ~40% organic matter. The statement at L429-431 is therefore not well-supported by the provided reference and I would suggest modifying or removing this statement.

We agree with the reviewer that the cited statement in Section 4.5 needed rephrasing. The hint at the thermal conductivity of soil minerals, including the chosen reference, may have been interpreted that the mineral content had been increased as a consequence of grazing. As shown in the results by Windirsch, this is not the case, instead the input of fresh organic material actually seems to have led to very high carbon content in the top soil at the most intensively grazed sites. We therefore changed the last sentence of this paragraph, formerly L.429 – 431, to "When this peat layer is trampled by herbivores, as observed at GR, the

soil thermal regime may be significantly modified, with effects on both thermal conductivity and diffusivity.", and removed the reference.

3. My earlier comment regarding soil and air temperature was mistakenly attributed to L408-410 in the previous version of the manuscript. It referred to the fact that the deep soil temperature difference (35cm) was on par with the observed difference in air temperatures (one degree, now mentioned at L248-249). Unless the difference in air temperature can be attributed to an effect of the grazing, it likely reflects spatial variability and observed differences in soil temperature smaller than this variability may not be relevant. This is now clearer with the changes to Fig1 paneling/y-axis and does not necessarily need addressing in the text.

OK, thanks a lot for the clarification.

L185: "Calculations were conducted using R" Please provide the version of R used. In addition, this should be moved either to the beginning or to the end of the Methods section, considering that R packages are already mentioned in earlier paragraphs.

The R-software is first mentioned at the end of Section 2.2 when referring to the R-package 'LakeMetabolizer'. We therefore moved the reference to the R-software to this place, rephrasing to "Calculations with this statistical package, as also for the other R-packages listed below, were conducted using R Version 4.1.1 (RCoreTeam, 2021)."

Figure 2: Please refer to Table B3 in the legend when mentioning pairwise comparisons. In addition, please double-check the post-hoc significance letters: I did not check them all but for instance it seems that for Reco GR1 should not share a letter with GR-3 according to Table B3.

According to the reviewer's suggestion, the first part of the caption of Figure 2 was changed to "Overview on C-fluxes at all chamber plots from July 7th to July 21st. Differing letters indicate significant differences between plots (p<0.01, please see also Table B3 for details). ..". The typos in the significance letters shown in Figure 2 have been corrected.

General comments

In this revised version of the manuscript, the authors excellently managed to clarify their study design for comparing carbon emissions and uptake between grazed and ungrazed Arctic tundra sites.

Detail additions in both introduction and methods will help readers to understand the study's intention and limitations. These limitations are picked up again in the discussion, and discussed in sufficient detail.

The additional work put into graphical design improves readability of the graphs and understanding of the "read thread" drastically.

Thanks a lot for this very positive evaluation!

Specific comments

Please consider making the data accessible via a scientific data repository.

Upon publication of this study, we plan to publish the data on the Open Research Data Repository EDMONT operated by the Max Planck Society (https://edmond.mpdl.mpg.de/imeji/).

Technical comments

Line 90: There's a missing space between 15 and km.

The space has been added.

Table 2: There is still some inequality in spacing of the asterisks in the table description.

We changed the formatting, so the asterisks now look even.

Line 741: There's a typo in the reference to Myers-Smith et al., where the I in "Macias-Fauria, M." should not be capitalized.

The respective reference has been corrected accordingly.